# Do psychiatric diseases follow annual cyclic seasonality?

**Hanxin Zhang**[1,2,3], **Atif Khan**[2], **Qi Chen**[4], **Henrik Larsson**[4,5], **Andrey Rzhetsky**[1,2,6]*

**1** Committee on Genetics, Genomics and Systems Biology, The University of Chicago, Chicago, Illinois, United States of America, **2** Department of Medicine, and Institute of Genomics and Systems Biology, The University of Chicago, Chicago, Illinois, United States of America, **3** Kansas City University, Kansas City, Missouri, United States of America, **4** Department of Medical Epidemiology and Biostatistics, Karolinska Institutet, Stockholm, Sweden, **5** School of Medical Sciences, Örebro University, Örebro, Sweden, **6** Department of Human Genetics and Committee on Quantitative Methods in Social, Behavioral, and Health Sciences, The University of Chicago, Chicago, Illinois, United States of America

\* andrey.rzhetsky@uchicago.edu

**Data Availability Statement:** Data can be obtained via licensing from IBM Health MarketScan (https://www.ibm.com/products/marketscan-research-databases). All data needed to evaluate the conclusions in the paper are present in the paper

## Abstract

Seasonal affective disorder (SAD) famously follows annual cycles, with incidence elevation in the fall and spring. Should some version of cyclic annual pattern be expected from other psychiatric disorders? Would annual cycles be similar for distinct psychiatric conditions? This study probes these questions using 2 very large datasets describing the health histories of 150 million unique U.S. citizens and the entire Swedish population. We performed 2 types of analysis, using "uncorrected" and "corrected" observations. The former analysis focused on counts of daily patient visits associated with each disease. The latter analysis instead looked at the proportion of disease-specific visits within the total volume of visits for a time interval. In the uncorrected analysis, we found that psychiatric disorders' annual patterns were remarkably similar across the studied diseases in both countries, with the magnitude of annual variation significantly higher in Sweden than in the United States for psychiatric, but not infectious diseases. In the corrected analysis, only 1 group of patients—11 to 20 years old—reproduced all regularities we observed for psychiatric disorders in the uncorrected analysis; the annual healthcare-seeking visit patterns associated with other age-groups changed drastically. Analogous analyses over infectious diseases were less divergent over these 2 types of computation. Comparing these 2 sets of results in the context of published psychiatric disorder seasonality studies, we tend to believe that our uncorrected results are more likely to capture the real trends, while the corrected results perhaps reflect mostly artifacts determined by dominantly fluctuating, health-seeking visits across a given year. However, the divergent results are ultimately inconclusive; thus, we present both sets of results unredacted, and, in the spirit of full disclosure, leave the verdict to the reader.

## Introduction

Psychiatric illness induces profound suffering and profoundly affects the lives of patients and their loved ones. Psychiatric disorders are special in the realm of complex diseases in that their

andits supporting information files. The source code and disease seasonality data for US can be accessed at https://github.com/hanxinzhang/seasonality. We also uploaded the data to the Dryad repository. The DOI is https://doi.org/10.5061/dryad.vdncjsxv6.

**Funding:** This work was funded by the DARPA Big Mechanism program under ARO contract W911NF1410333 (AR), by National Institutes of Health grants R01HL122712 (AR), 1P50MH094267 (AR), and U01HL108634-01 (AR), and by a gift from Liz and Kent Dauten (AR). The funders had no role in study design, data collection and analysis, decision to publish, or preparation of the manuscript.

**Competing interests:** The authors have declared that no competing interests exist.

**Abbreviations:** ADHD, attention-deficit/hyperactivity disorder; AK, Alaska; AWMN, Alaska, Washington, Montana, North Dakota; DR, diagnosis rate; EM, expectation–maximization; FL, Florida; IRB, institutional review board; MCMC, Markov chain Monte Carlo; ME, Maine; MT, Montana; ND, North Dakota; SAD, seasonal affective disorder; TX, Texas; UTI, urinary tract infection; WA, Washington; WAIC, Watanabe–Akaike information criteria.

diagnoses almost exclusively rely on outwardly subjective symptoms, presented by the patient and interpreted by a psychiatrist. On the other hand, infectious and mendelian diseases occupy space in the diagnostic continuum's highest-certainty extreme, which can typically be ascertained definitively via specialized experimental tests. Supporting this view of etiologic entanglement, psychiatric disorders appear to share extensive genetic and environmental predispositions. For example, whole-genome association data [1–4] analysis indicates that psychiatric disorders are highly genetically correlated, while large-scale, family-based studies, supporting these highly genetic correlations across psychiatric maladies, also suggest that these disorders possess shared environmental risk factors [5,6]. The estimated shared proportion of environmental risk factors between nonpsychiatric complex diseases, and within psychiatric and nonpsychiatric disease pairs, tend to be much lower [5,6].

If psychiatric disorders share many environmental risk factors, it should be possible to identify common environmental stimuli affecting many [7]—or even all—of them. One of the potential environmental drivers of selected psychiatric conditions, such as seasonal affective disorder (SAD) [8–11] and depression [12–17], are the annual and daily sunlight cycles, which drive the circadian clock. Both SAD and depression tend to worsen during darker seasons. It is unclear whether this seasonal pattern is shared by other psychiatric disorders and whether this disease seasonality is solely limited to particular geographic areas. This study's main hypothesis is that the bulk of psychiatric maladies share this annual light dependency cycle. Here, we systematically examine psychiatric conditions vis-à-vis their annual cycle of disorder-specific patient visits, as represented in clinical records, across very different geographic zones and 2 distinct continents, Europe and North America. For reference, we compare annual psychiatric disorders' reporting cycles with those for infectious disease, across U.S. citizen and Swedish populations.

The ideal data input required to answer our questions about disease seasonality would include records of direct, physician-led patient health state evaluations, following patients over many years, directly detecting their health state improvement or deterioration. Unfortunately, such data are yet to be generated. Instead, we used very large collections of electronic medical records, documenting patients' visits to medical practitioners, along with diagnoses, procedures, and prescribed medications. The latter type of data is subject to biases, such as weather events (think of blizzards), holidays, and vacations, all of which affect the behavior of both doctors and patients. To account for both these biases and for possible noise in data, we developed a family of statistical models, estimating annual disease diagnosis rates' (DRs) most likely seasonal oscillation patterns, while striving to account for data biases. We then tested these models against the data to determine the best model that did not overfit observations.

Our study used the IBM Watson Health MarketScan dataset [18] containing insurance claim records of over 150 million of unique U.S. citizens and the Swedish National Health Register [19] detailing the health dynamics of virtually all Swedes, with over 11 million unique people visible in the data. The US data cover the time interval between 2003 and 2014, while the Swedish data encompass an interval between 1980 and 2013. Although the IBM MarketScan database is one of the largest and most comprehensive collections of US insurance claims, it was built by merging asynchronous subsets collected by multiple private health insurers. As a result, the data have layers of idiosyncratic properties that complicated our analysis (Fig 1). To account for systematic biases and noise in data, we designed a multilevel Bayesian model describing the generation of the observed disease-specific patient visit counts (see Fig 2 and the Materials and methods section for details). We present results from 2 distinct analysis approaches, the first with "uncorrected" counts of diseases-specific visits and the second with "corrected" seasonality that we adjusted for seasonal changes in all-cause medical visits.

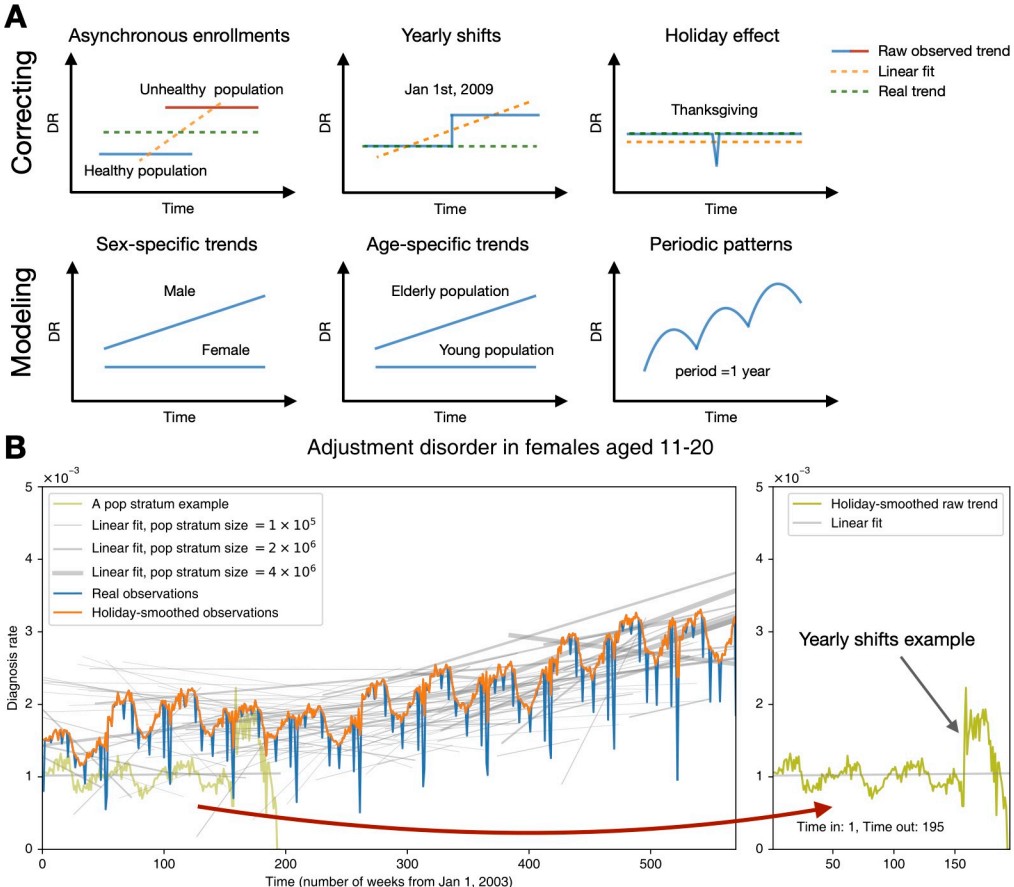

**Fig 1. The characteristics of the US data and how it influences our model design. (A)** Our modeling aimed to correct biases and noise in the MarketScan database and to infer a latent disease DR trend and seasonality for specific age–sex groups. The upper left panel describes the scenario in which 2 populations—a healthy one (the blue line) and an unhealthy one (the red line)—enrolled in the data at different times. The blue and red lines represent the trend of the DR for the 2 populations. We can see that the healthy population joined and left our data earlier than the unhealthy one. Thus, if we fit a simple linear regression model, the result may lead us to conclude that there is an upward trend of DR. Nonetheless, the real trend is actually constant if we had the ability to collect the data of all time (synchronous enrollment) for both populations. The trend of the linear fit (the orange line) comes from "asynchronous enrollments" of populations in various health statuses. The rest of the panels delineate other scenarios likewise. **(B)** This subplot shows the overall trend and seasonality for a sample disease. The holiday-smooth function offsets the effect of holidays and celebrations that decrease the DR sharply (the orange curve vs. the blue curve). Bear in mind that patients joined and left our US data asynchronously. The gray lines illustrate varying linear fit trends of population strata, defined according to their enrollment dates (see the Methods and techniques part of the Materials and methods section). Some population strata include more people, while others are smaller in size, as marked by the gray lines' different widths. A sample population stratum enrolled from week 1 to 195 is highlighted in the right panel. Notice that the sudden shift still exists—even for a population with a consistent composition, meaning that the shifts do not result from enrollment changes. The data underlying this figure can be found in https://doi.org/10.5061/dryad.vdncjsxv6. DR, diagnosis rate.

## Bayesian model summary

Fig 1B illustrates a typical disease's overall trend and seasonality, summarizing divergent linear trends of population strata (patient cohorts with the same entry and exit points within our database, represented by the gray lines). The real observation curve was calculated by dividing the total diagnoses by the total enrollees (DR) at each time point (in this study, by week). The holiday-smooth function uses the average DRs around known holidays to calculate and offset the sharp decrease in the DR shown around holidays and other celebrations.

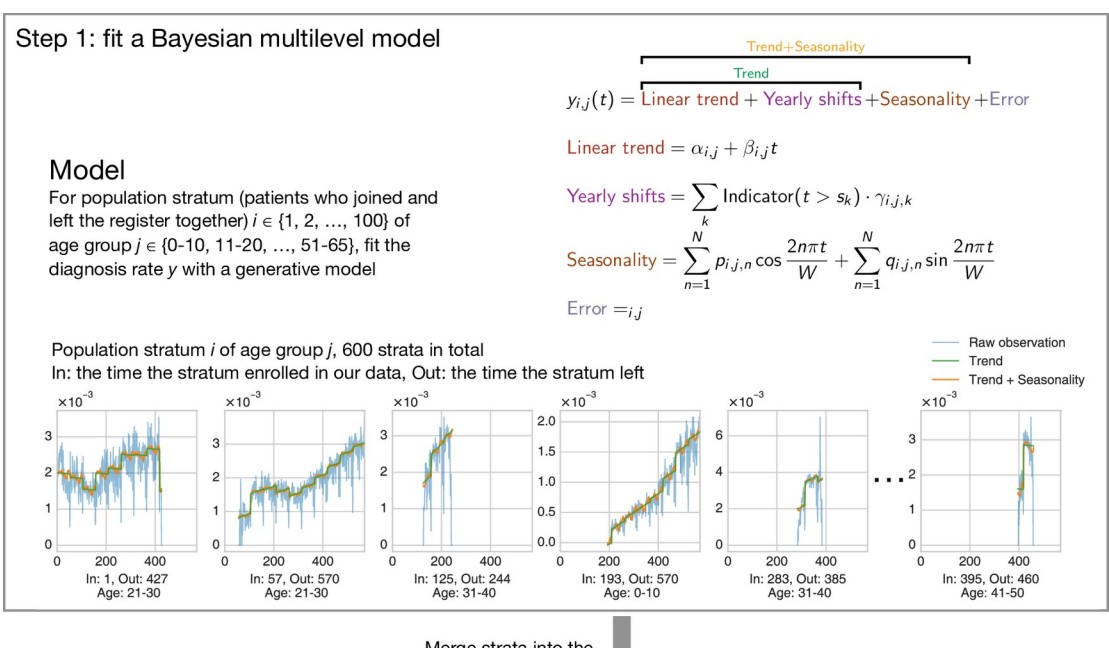

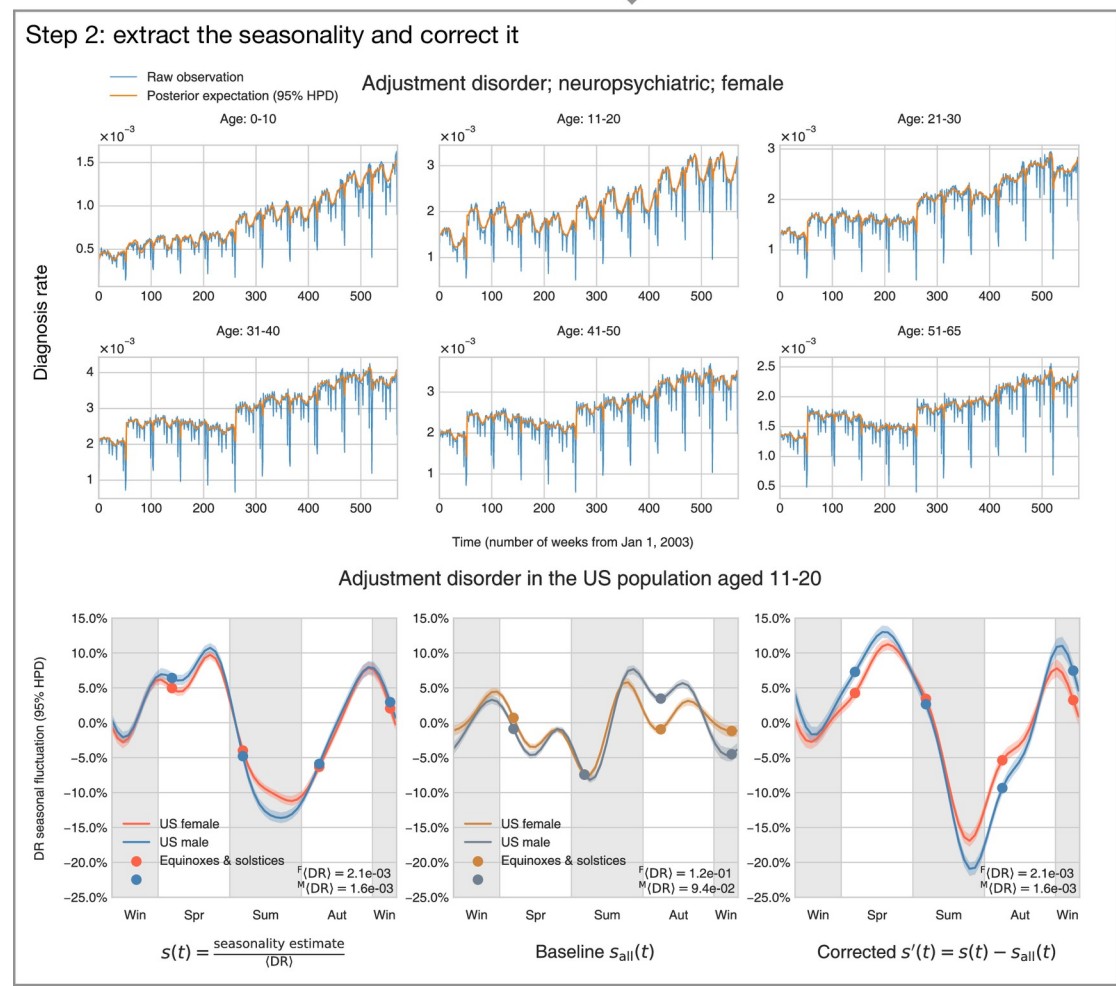

**Fig 2. The method and procedure to infer seasonality. Upper frame (Step 1):** We modeled the DR by decomposing it into several parts: the linear trend, yearly shifts, seasonality, and an error term. We assigned people into hundreds of population strata according to their enrollment dates. A total of 6 populations of specific age and enrollment dates are shown here. The model fits each population strata separately (but not independently), with shared priors and hyperpriors so that information can be shared across populations. "In" means the time the stratum joined our data (enrollment beginning), and "Out" indicates the time they left (enrollment end). **Lower frame (Step 2):** After obtaining the estimates of all model parameters, we were able to extract the seasonality and make inferences. The upper plot shows that the posterior expectation (mean) reproduces our raw observation very well, which partly validates that our model is mixing well. Note that, for this particular condition, the 95% highest posterior density interval is very small, so it may be difficult to tell it in the plot (light green shade). The left subplot at the bottom exemplifies how we can find the relative seasonal fluctuation (uncorrected) $s(t)$ by dividing the seasonality estimates by the time-average DR ($\langle DR \rangle$, Expression (17)). We can possibly correct for the baseline fluctuation of all medical visits by deducting the $s_{all}(t)$ (representing the uncorrected seasonality of all medical visits) from $s(t)$ and obtain the corrected seasonality $s'(t)$ (right subplot at the bottom). The data underlying this figure can be found in https://doi.org/10.5061/dryad.vdncjsxv6. DR, diagnosis rate.

The gray lines (Fig 1) show the population strata's linear fit trends enrolled in the data asynchronously. For example, the olive curve represents a group of patients enrolled from week 1 to week 195. Different population strata do not show a uniform trend—some gray lines go upward, some are flat, and some go downward. Fig 1B demonstrates that, due to heterogeneous insurance enrolling practices, groups of people joining an insurer together do not resemble a random sample from the general US population. In addition to "asynchronous enrollment," we also found that many diseases' DRs suddenly shifted at the beginning of every year for many population strata of consistent composition. The sample population stratum could give us an idea of such shifts (shown in the olive curve on the right panel of Fig 1B).

We designed a multilevel Bayesian model to describe the generation of the observed DR, given several sources of systematic bias and noise (Fig 2). First, we grouped patients based on their ages and enrollment dates and defined "population strata," which are cohorts containing patients of the same age-group and enrollment date in our data in the same time interval. We then modeled each population stratum's trend and seasonality separately, but not independently. We shared the information across age-groups and population stratum because they were sampled from the same priors and hyper priors. For example, for the linear trend intercepts $\alpha_{i,j}$ for population stratum $i$, we sampled them from a skew normal distribution with an age-specific center $\mu_j^{\alpha}$, scale $\sigma_j^{\alpha}$, and shape $h_j^{\alpha}$ (Fig 2, Step 1). These age-specific hyperparameters were also sampled from shared Gaussian process hyperpriors that chained them together across age-groups so that close ages would have close center $\mu_j^{\alpha}$, scale $\sigma_j^{\alpha}$, and shape $h_j^{\alpha}$ (see the Materials and methods section for more information).

We estimated all parameters simultaneously using a Markov chain Monte Carlo (MCMC) sampler [20]. After obtaining all the estimates for every population stratum, we can merge strata and find the age- or sex-specific trends and seasonalities, as shown in the top panel of Fig 2, Step 2. We highlighted a yearly seasonality sample in the bottom panel of Fig 2, Step 2. Fig 2, Step 2's lower left plot gives the relative seasonal fluctuation of a sample disease, computed by dividing the raw seasonality estimate by the time average of observed DR (see Expressions (17) and (18) of Materials and methods).

In an attempt to account for season estimates' possible non-biology–driven fluctuations (vacations, bad weather, and holidays), we attempted normalizing the raw DR using the DR of all medical visits (shown in the lower center panel of Fig 2, Step 2 and S1 Fig). The resulting corrected seasonality then represented the count excess/deficit with respect to the baseline medical diagnoses fluctuation (the lower right panel of Fig 2, Step 2). In the present work, we refer to the uncorrected seasonality relative to the time-average DR as "uncorrected" seasonality or "$s(t)$." We refer to the seasonality corrected by the all medical visits baseline as "corrected" seasonality or "$s'(t)$."

## Results

We applied our statistical models to probe the annual seasonality of 33 psychiatric and 47 infectious diseases in 2 sexes and multiple age-groups. For simplicity of visualization and discussion, we used the meteorological season conventions, defined as follows: winter starts on December 1 and ends on February 28 or 29, spring starts on March 1 and ends on May 31, summer starts on June 1 and ends on August 31, and autumn is the rest of the year. In this description, we focus on the results for the 5 most prevalent psychiatric disorders and the 5 most common infectious diseases, but the results for all the diseases studied, using both corrected and uncorrected seasonality analyses, are available in S1–S10 Data (results data split into 10 files). The data can also be found on the project repository at https://github.com/hanxinzhang/seasonality.

### Uncorrected seasonality analysis

We first analyzed diseases' uncorrected seasonalities without considering the underlying baseline fluctuation of all medical visits. Psychiatric disorders appear to follow a nearly identical yearly cycle of care access patterns; on average, they spike in the darker periods and recede during warmer and brighter times (see Fig 3 for uncorrected seasonality), although all patterns were exceedingly more complicated than a unimodal curve. Fig 3 shows disorder-, sex-, and age-group–specific seasonalities for the 5 most diagnosed psychiatric conditions in the US: depression, anxiety/phobic disorder, adjustment disorder, substance abuse, and attention-deficit/hyperactivity disorder (ADHD). We show matching results for Sweden, time aligned with their US counterparts, but scaled by 0.3 in magnitude for ease of comparison. Clearly, despite significant differences in social, economic, cultural, and healthcare management of these conditions in the 2 countries, the curves are surprisingly similar across both countries for the same conditions and highly consistent across the disorders. The plots are designed to show deviation from the yearly mean value in percent of disorder-specific visits at given time points. A uniform pattern of visits through the year would result in a flat line at 0%. In the plots, we see around a 10% to 20% fluctuation relative to the yearly mean in the US. Seasonal fluctuations in Sweden are even larger, reaching a 70% decrease in patient visits related to, for example, ADHD. For all 5 most prevalent conditions, especially in the US, people younger than 20 seem to experience larger-scale seasonal variation in psychiatric visit frequency. In Sweden, however, the difference in seasonal variations across age-groups is minor. In terms of the discrepancy between the 2 sexes, females and males bear analogous seasonality in both countries. It is worth mentioning that ADHD in people older than 20 demonstrates a distinctive seasonality that rises gradually from autumn to winter.

Looking only at psychiatric disorder results, one might conjecture that the observed annual regularities are common for all diseases and that the cycle dynamics are mainly driven by social factors. This is far from being true, as shown in the annual infectious disease cycles (Figs 4 and 5). A low-dimensional embedding of estimated seasonality harmonics using the Isomap algorithm [21] (Fig 5, https://seasonality-web-app.herokuapp.com) reveals that psychiatric curve shapes are tightly clustered (similar), while curve shapes for infectious diseases are very diverse, and, therefore, scattered in the embedding representation.

If we consider the 5 most diagnosed infectious diseases in the US (acute upper respiratory infection, ear infection, acute bronchitis, urinary tract infection (UTI), and cellulitis, see Fig 4), the patterns are very different. The magnitudes of seasonal variation are comparable between the US and Sweden for infectious diseases, so the curves are scaled in the same way. As expected, in the US diagnoses, the 2 respiratory infections (acute upper respiratory infection and bronchitis) rise in colder times, peaking in the early spring, and subside in warmer

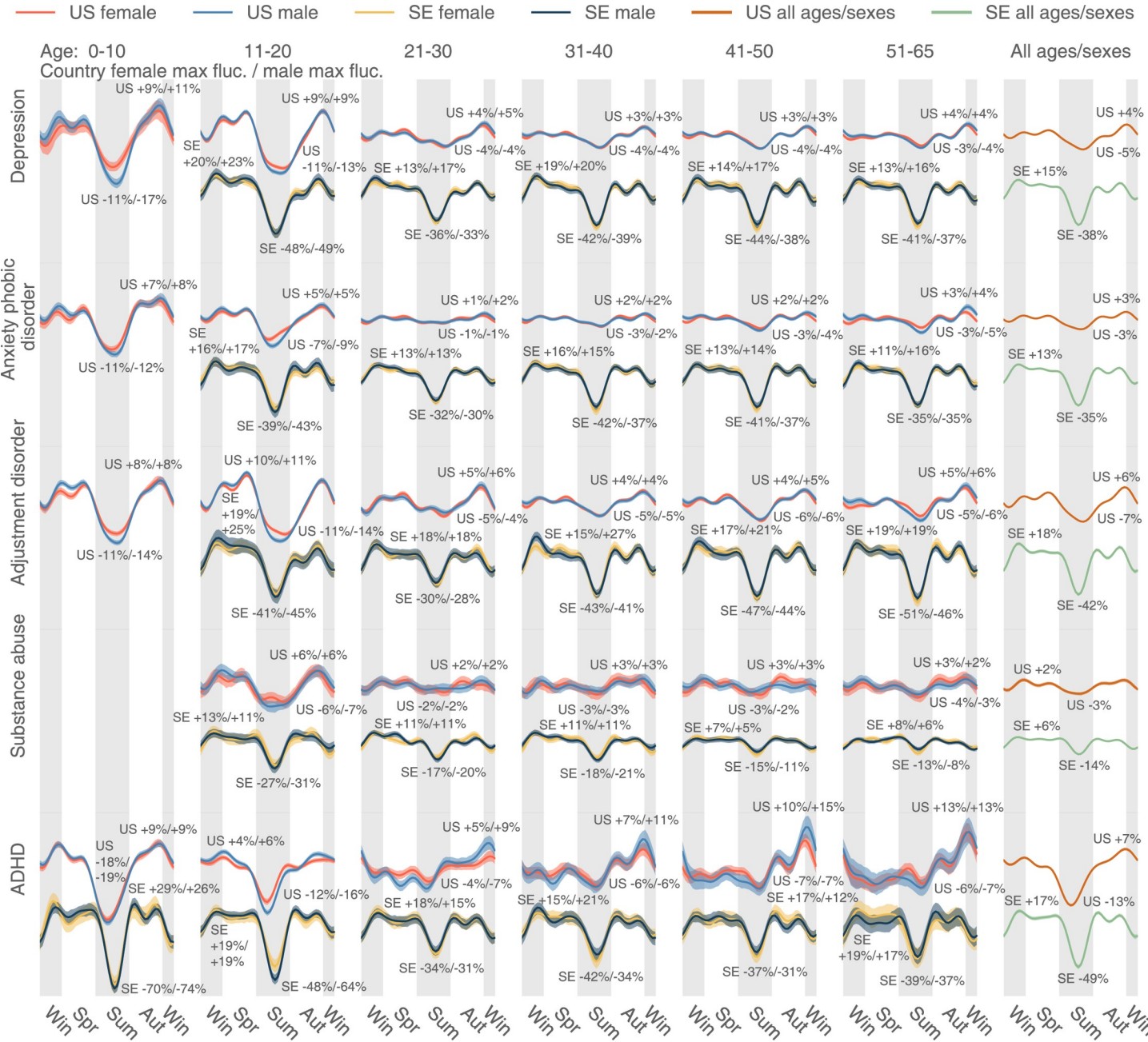

**Fig 3. The uncorrected seasonality plots of the 5 most diagnosed psychiatric diseases in the US: Depression, anxiety/phobic disorder, adjustment disorder, substance abuse, and ADHD.** The results in SE are juxtaposed, but scaled by 0.3 in magnitude for clearer comparison. We plotted all lines based on a weekly DR estimated as the total number of diagnoses in a week, divided by the total number of enrollees in our database in the week. Positive and negative maximum fluctuations compared to the mean DR are text-labeled following a format: Country female maximum fluctuation in percentage / Male max fluctuation in percentage. We use the meteorological seasons defined as follows: Winter starts from December 1 and ends on February 28, spring starts from March 1 and ends on May 31, summer starts from June 1 and ends on August 31, and autumn is the rest of the year. We discarded the health records of people over 65 because the majority of that population in the US data switched to Medicare, and remaining records were not representative. A disease could be extremely rare in some age–sex brackets. The plot only shows those age–sex–specific seasonalities with a time-average DR (Expression (17)) larger than $1 \times 10^{-5}$. The data underlying this figure can be found in https://doi.org/10.5061/dryad.vdncjsxv6. ADHD, attention-deficit/hyperactivity disorder; DR, diagnosis rate; SE, Sweden.

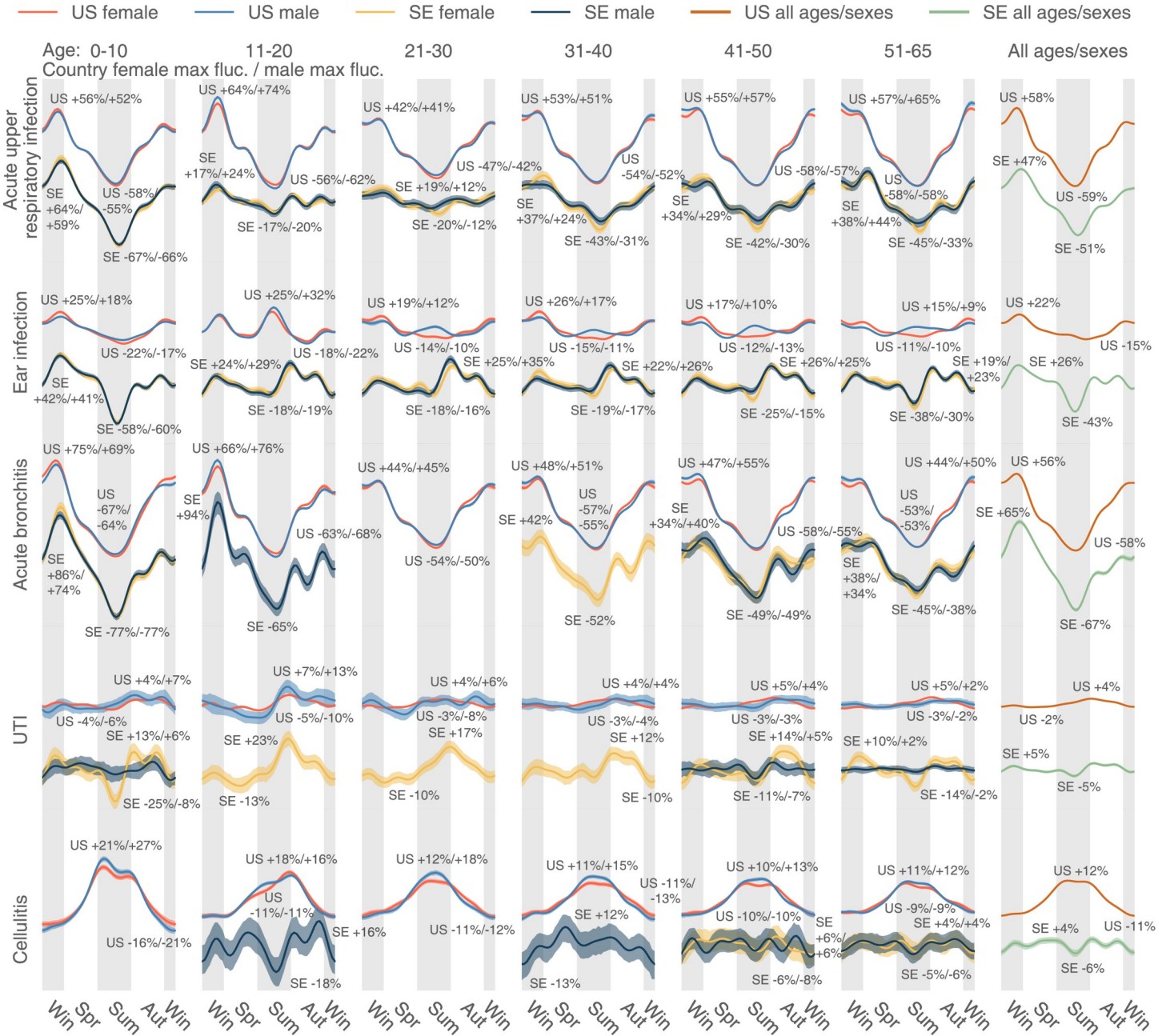

**Fig 4. The uncorrected seasonality plots of the 5 most diagnosed infectious diseases in the US: Acute upper respiratory infection, ear infection, acute bronchitis, UTI, and cellulitis.** The results in SE are juxtaposed without scaling. A disease could be extremely rare in some age–sex brackets. The plot only shows those age–sex–specific seasonalities with a time-average DR larger than $1 \times 10^{-5}$. The data underlying this figure can be found in https://doi.org/10.5061/dryad.vdncjsxv6. DR, diagnosis rate; SE, Sweden; UTI, urinary tract infection.

days, with the lowest rate at the end of summer. On the contrary, cellulitis, a deep skin infection, rises in warmer periods and subsides in the winter in the US—similar to general skin infections (S2 Fig). In Sweden, cellulitis is extremely rare in children and young females; in males and older adults (over 40 years old), it shows no obvious patterns, possibly because cases of this disease are sparse in this northern and relatively small country.

**Fig 5. The embedding of uncorrected seasonality curves in a low-dimensional space suggests the homogeneity of the psychiatric diseases' seasonal variation.** We used the Isomap method to obtain a low-dimensional seasonality embedding of the first 10 Fourier harmonic base estimates $\bar{p}_{j,1}, \bar{p}_{j,2}, \ldots, \bar{p}_{j,5}, \bar{q}_{j,1}, \bar{q}_{j,2}, \ldots, \bar{q}_{j,5}$ (see Expressions (15) and (16)). Compared to the infectious diseases, we can see that the embeddings of psychiatric disease harmonics concentrate in a smaller space, implying the relative homogeneity of their seasonality. The data underlying this figure can be found in https://doi.org/10.5061/dryad.vdncjsxv6.

Ear infections in children (newborns to 10 years old) are more common in winter and less common in summer in both countries, as expected (Fig 4). Unlike psychiatric disorder trends, we discern a distinct "peak triplet" pattern of ear infection in US teenagers (11 to 20 years old) with high DRs in both the summer and winter and low DRs in spring and autumn. This "triplet" pattern extends to older US age-groups and is visible with considerable variation in the Swedish cohort. Finally, UTI seasonality in the US tends to be level except in teenagers, but it grows from summer to autumn and goes down in winter and spring in Sweden, particularly in females aged between 11 and 40 years (Fig 4).

We conducted an additional analysis over the MarketScan data to probe the seasonal variation differences among higher- and lower-latitude geographic regions. First, we conducted separate analyses using data exclusively representing the 4 high-latitude states in the US: Alaska (AK), Washington (WA), Montana (MT), and North Dakota (ND). We did not include Maine (ME) due to its relatively lower latitude (Portland, ME 43.7˚ N versus Seattle, WA 47.6˚ N) as compared to the selected 4 states. We found that all 5 most prevalent psychiatric disorders demonstrate larger seasonal oscillation in the 4 high-latitude states (AK, WA, MT, and ND, abbreviated in this study as AWMN) than in the whole country (S3 Fig). For example, in the summer, depression goes down about 23% in females aged 11 to 20 in each of the 4 states, contrasted to an 11% decrease for the country on average for the same group. In 11- to 20-year-old males, ADHD decreases by 16% in the whole country, but 26% in the 4 high-

latitude states. In general, the fluctuation magnitude is around 1.5 to 2 times larger in AK, WA, MT, and ND, but it is still much smaller than the variation in Sweden, which is at an even more northern latitude. Second, we observed that, for infectious diseases, the magnitude of seasonal variation was similar between the whole country and the 4 high-latitude states (S4 Fig). We then examined 2 large states in the South: Texas (TX) and Florida (FL). We did not include other southern states, such as Hawaii or Louisiana, due to the smaller population size represented in our data. Louisiana and other continental southern states are also not as south in latitude as TX and FL. Likewise, we did not consider California because a large part of it spans more northern areas. For either psychiatric disorders or infections, the results are similar to those of the whole US (S5 and S6 Figs). It is remarkable that for psychiatric conditions such as ADHD in males aged zero to ten and 11 to 20, the variation in TX and FL is smaller. We saw this seemingly smaller variation tendency in other psychiatric disorders as well, but the variation is not as significant as the comparison between the US and Sweden or between the US and the 4 high-latitude US states.

To summarize, we observed a consistent, seasonal pattern in psychiatric disorders, with a shared recess in the summer, as well as a shared increase in the fall, in both the US and Sweden (Fig 3). Diverging from the conclusions of a smaller-scale earlier study, which found only limited seasonal changes in general mental disorders [22], we observed that seasonality is shared by a large number of psychiatric disorders—in spite of their diverse symptomatology and prevalence. In addition to observing the abovementioned seasonality in depression, anxiety, and adjustment disorders (Fig 3), we detected similar patterns in many other psychiatric disorders such as schizophrenia and related psychoses (S7 Fig) and migraine (S8 Fig). By contrast, we found heterogenous seasonality patterns across infectious diseases.

## Corrected seasonality analysis

While computing corrected seasonality plots, we grouped different age-group curves together on the same subplots to compare the variability of seasonality across ages. We set the y-axis limits to be identical (for the same geographic area), so it is easier to compare seasonality across diseases. For each analyzed disease, we also gave its overall seasonality, aggregating all ages and sexes (the third and sixth columns of Figs 6 and 7 and S9–S12 Figs). The time-average DR ⟨DR⟩ on the plots indicates disease prevalence in a particular sex–age bracket, and it could suggest what subpopulations are the most representative groups for a disease.

For the 5 most diagnosed psychiatric disorders in our US data (Fig 6), most sex–age groups' seasonality flattened after correcting for the baseline fluctuation of all medical visits. US patients aged 11 to 20 are exceptional and still show an evident DR decrease in the summer and upward trends in the spring, autumn, and winter after adjusting for the baseline seasonality of all medical visits. Depression, for example, decreases 20% more than the all medical visit baseline in the summer for both females and males aged 11 to 20 in the US. The age–sex aggregated curves do not suggest much seasonal variation for depression, anxiety phobic disorder, adjustment disorder, and substance abuse in the US, as the age 11 to 20 group is not dominant in terms of disease prevalence. By contrast, for ADHD, the population aged 11 to 20 is the most representative, so we can observe that the summer's seasonal decrease in this condition in the age–sex aggregated plot shown in the third column of Fig 6.

In Sweden, the correction strongly adjusted the observed seasonality in psychiatric disorders (Fig 3) because the baseline variation of all medical visits is large (the fourth row of S1 Fig). After correction, we found only a minor decreasing trend in the summer for depression in 11- to 20-year-old patients. It seems that DR for substance abuse goes up in the summer in Sweden, opposite to the trend in the US. Note that before applying the baseline correction, the

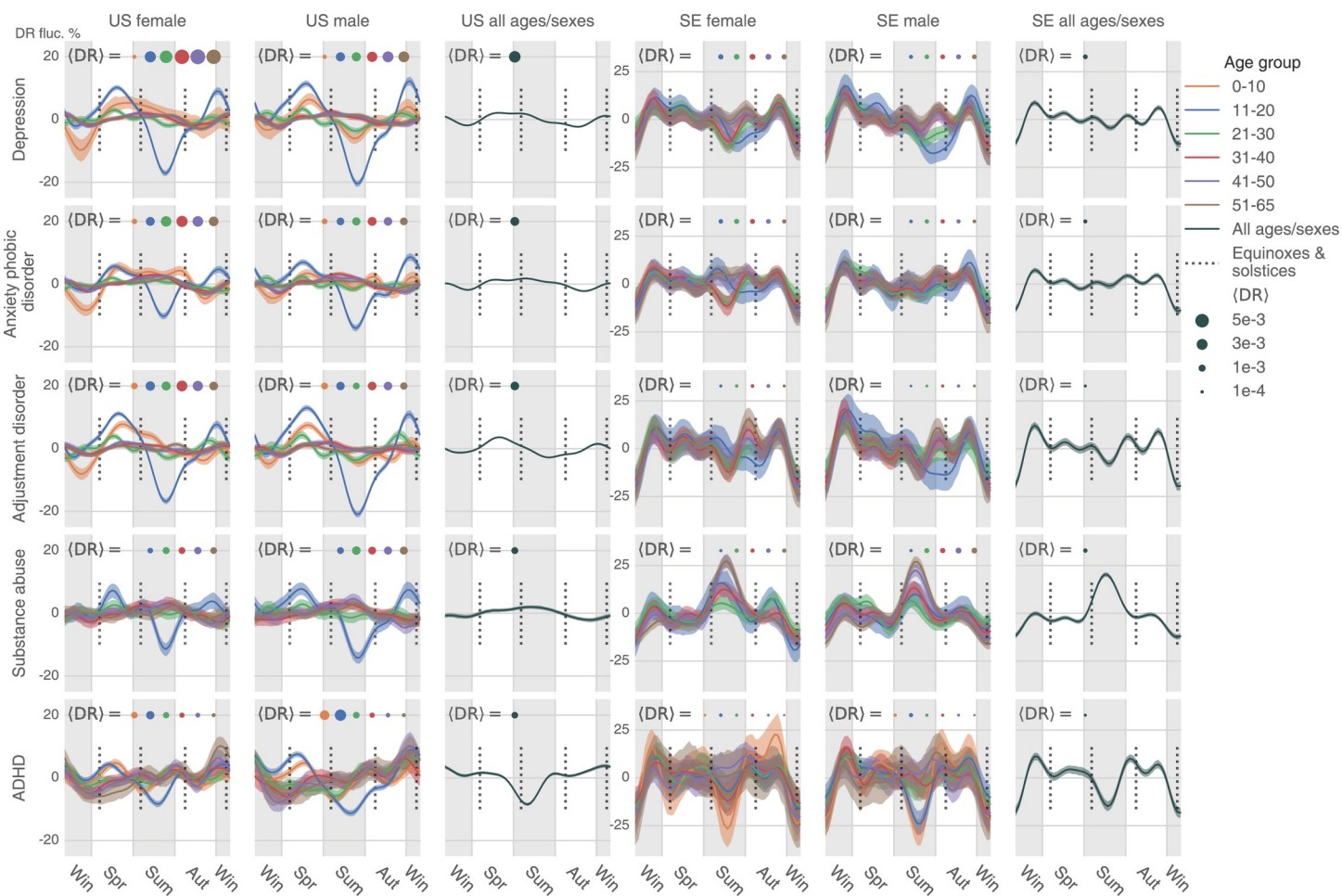

**Fig 6. The corrected seasonality plots of the 5 most diagnosed psychiatric diseases in the US and SE: Depression, anxiety and phobic disorder, adjustment disorder, substance abuse, and ADHD.** ADHD, attention-deficit/hyperactivity disorder; DR, diagnosis rate; SE, Sweden.

substance abuse DR decreases in the summer (Fig 3). Therefore, the peak in the Swedish summer only suggests that such a seasonal decrease in substance abuse does not exceed the baseline variation. Besides, in Sweden, we also observed a decreasing DR in ADHD in the summer. Finally, for psychiatric disorders, we noticed that there is a uniform decrease in DR at the beginning and end of the year (Figs 3 and 6), possibly due to winter break or vacation, which is even more obvious in Sweden.

After correction, the 5 most diagnosed infectious diseases in the US maintain significant seasonality (Fig 7), almost consistent across age-groups, sexes, and 2 countries (the US and Sweden). The seasonal trends are comparable to the uncorrected trends (Fig 4). In the summer, we found decreased DR for acute upper respiratory infection and increased DR for cellulitis. The distinct peak in summer ear infections still exists for US teenagers (11 to 20 years old) and some older groups in Sweden.

Additionally, we studied the seasonal variation differences across higher- and lower-latitude regions after correction (S9–S12 Figs). Similar to what we found in the uncorrected analysis, psychiatric disorders in 11 to 20 year olds demonstrate larger-than-national-average seasonal oscillation in the 4 high-latitude states (AK, WA, MT, and ND or AWMN, S9 Fig) and smaller-than-national-average seasonal oscillation in 2 southern states (TX and FL, S11 Fig).

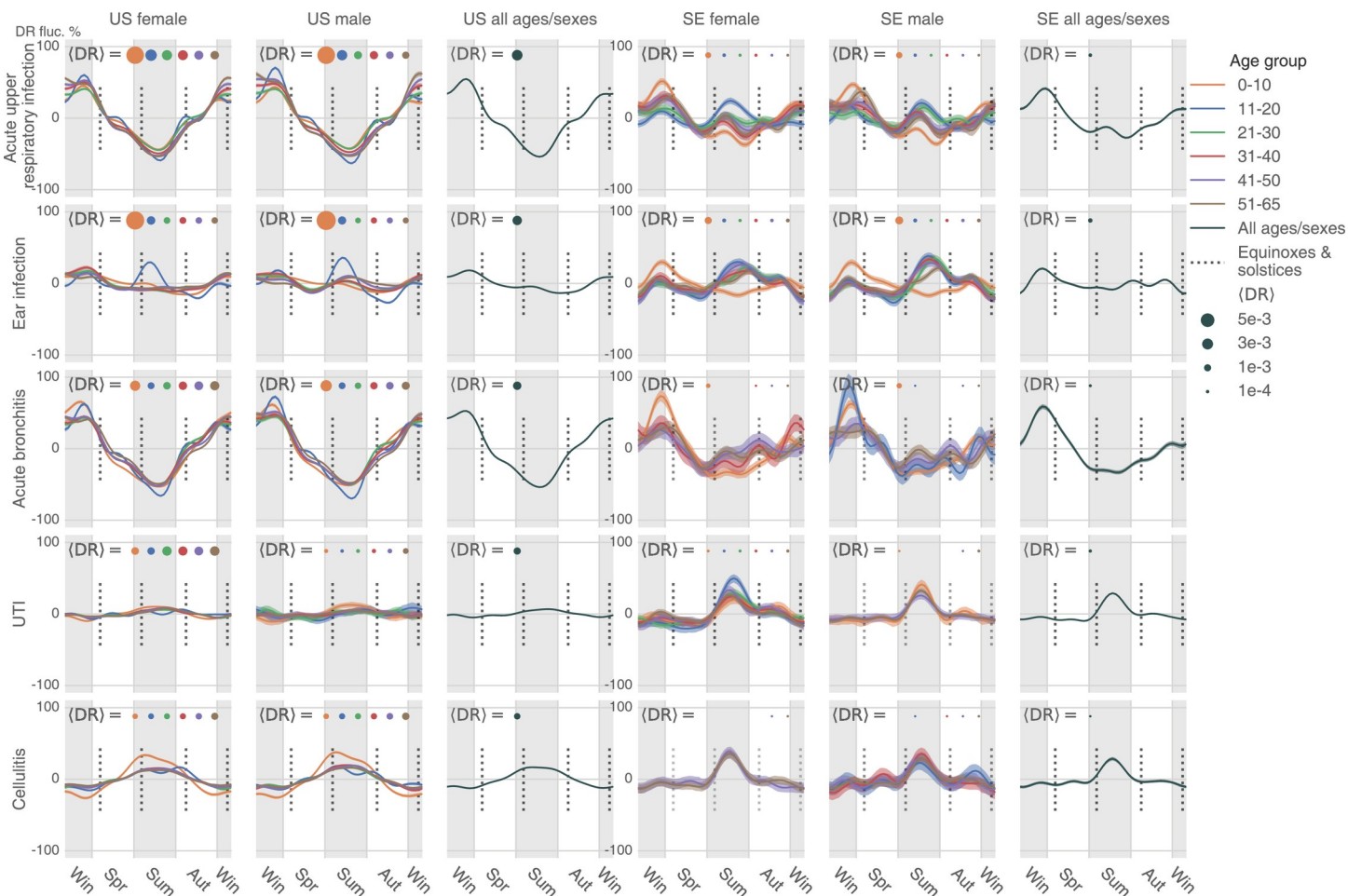

**Fig 7. The corrected seasonality plots of the 5 most diagnosed infectious diseases in the US and SE: acute upper respiratory infection, ear infection, acute bronchitis, UTI, and cellulitis.** The data underlying this figure can be found in https://doi.org/10.5061/dryad.vdncjsxv6. DR, diagnosis rate; SE, Sweden; UTI, urinary tract infection.

Fig 8 merges all psychiatric disorders in 11 to 20 year olds and shows the seasonal fluctuation differences across the 4 most northern states (AWMN, largest fluctuation), the whole US (middle), and 2 southern states (smallest fluctuation).

## Discussion

An infectious disease, in its acute manifestation, requires the immediate attention of a physician. Most infectious diseases are transient (aside from a few that are chronic, such as malaria, AIDS, and herpes). Therefore, we expect that annual encounter rates of healthcare-seeking patients suffering with infections reflect real seasonality rather than exclusively annual patterns of recreational activities and vacations.

The situation might be different with patient visits associated with care for chronic diseases, elective procedures, and routine health checkups; harsh weather and vacation time may delay a visit and may require adjustment in analysis. Therefore, we produced 2 versions of analysis of annual disease-related visit rates: "uncorrected" and "corrected." The former type of analysis answers the question "How are healthcare-seeking visits of patients with disorder X distributed across seasons?" The latter type of analysis answers a different question: "How are the

## All psychiatric disorders in 11-20 years

**Fig 8. The corrected seasonality of all psychiatric disorders in 11 to 20 year olds across 4 regions.** In the US, the annual oscillation of psychiatric disease DR is larger in the high-latitude areas (AK, WA, MT, ND, or AWMN) than in the low-latitude areas (TX and FL). The data underlying this figure can be found in https://doi.org/10.5061/dryad. vdncjsxv6. AK, Alaska; AWMN, Alaska, Washington, Montana, North Dakota; DR, diagnosis rate; MT, Montana; ND, North Dakota; WA, Washington.

proportion of healthcare-seeking visits of patients with disorder X distributed across seasons with respect to all healthcare-seeking visits for this time interval?"

First, we argue that the psychiatric conditions we chose to examine in this study behave more like acute infections rather than elective procedures. For example, depression cases recorded in the Swedish registries are the most severe cases, where the patient needs immediate hospitalization (for example, because a patient stops eating). Milder depression cases, where a patient requires an antidepressant prescription, are handled by primary care providers and do not show up in health registries.

Second, our uncorrected seasonality results better resonate with existing published, smaller-scale observations than our corrected results: Past studies report anxiety and depression rates as higher during the spring than in the summer in both Europe [23] and the US [24]; Bipolar disorder symptoms receded in the summer in patients in Arctic areas of Northern Fennoscandia [25], and; depression and suicide in the US were reported to be higher in the spring than in the summer [26]. Past studies also reported an increase in substance abuse–related admissions to hospitals in the spring, as compared to the summer in Vietnam [27]. Studies in Vietnam have also reported that mood disorder–related hospital admissions were higher in the spring and fall compared to the summer, in agreement with our uncorrected

seasonality results [27]; in the corrected results, a subset of age-groups displayed higher rates of depression in the summer than in the fall, as shown in Figs 3 and 6.

Third, our attempt to account for the seasonality of individual diseases rates by adjusting for the total rate of healthcare-seeking events (see Figs 6 and 7) produced results somewhat discordant with the findings of previous published studies. For example, while uncorrected substance abuse rates in Sweden [28,29] have been reported as higher in the spring than in the summer (and this is what our uncorrected seasonality data show), in the corrected Swedish results, the substance abuse rates completely reversed this trend, with the disorder rates higher in the summer than in the spring. In the US, corrected data also changed drastically with respect to the uncorrected—for psychiatric conditions only 1 age-group, 11- to 20-year-old patients, preserved their behavior across all psychiatric conditions compared to the uncorrected seasonality results. The rest of the age-groups acquired additional idiosyncratic properties that are not aligned with anything known about these diseases. The results suggest that correcting by all visits may not be optimal because the seasonality of all visits does not reflect nonspecific, health-seeking behaviors. The evidence shows that this is more likely determined by the seasonality of the dominant acute diseases (such as infections).

Our uncorrected version of this study suggests the existence of a uniform seasonality in the psychiatric disorder DR. This reported regularity was discovered via an analysis of a very large volume of health data, eliminating the possibility of noise-driven, spurious results. However, interpreting these statistically stable trends requires caution. The evidence that seasonal patterns for other psychiatric disorders closely follow those for SAD and depression does suggests a plausible link to the annual daylight cycle, in turn affecting human circadian rhythms; yet, we cannot completely rule out the influence of societal and economic factors. Furthermore, the other causality direction cannot be eluded at this stage: that psychiatric symptomatology due to light/dark cycle changes may lead to decreased social activity.

Most importantly, our analyses were based on the interpretation of diagnostic code time stamps, entered by physicians or psychiatrists after professional visits with patients. Here, we implicitly interpret the frequent psychiatric visits of a number of patients at the same time interval as evidence of the population's deteriorating mental health. (This assumption is reasonable with infectious diseases, requiring a doctor's immediate attention after symptoms manifest themselves.) One may argue that lower psychiatric diagnostic rates in the summer are caused by vacations taken by either the patient or the psychiatrist and not by the disease itself. This explanation is made less likely by the replication of the same annual disease registration pattern across different latitudes in the US and Sweden—because the vacation cultures of the 2 countries (and of the North and the South of the US) are drastically dissimilar; US vacations are typically shorter than their Swedish counterparts and tend to occur asynchronously around the year.

Past psychiatric seasonality studies [30,31] used relatively small cohorts, typically insufficient to distinguish seasonal variation in disease prevalence from the background noise—with a few important exceptions. SAD is the most recognized, seasonality-related psychiatric condition, a subtype of a unipolar depression [12–17]. Etiological hypotheses and experimental data have connected SAD to human circadian rhythms, the daily duration of exposure to sunlight, a patient's individual genetic variation, and their neurotransmitters' biochemistry [8–11]. Another plausible mechanism of seasonal changes in psychiatric conditions is seasonal immune dysregulation due to seasonal allergies and infections [32]. We observed that anxiety and phobic disorders follow annual cycles nearly identical to those of SAD and depression—contrary to earlier studies' conclusions [30,31] regarding anxiety's lack of seasonality. Previous examination of anxiety in smaller cohorts in the Netherlands found virtually no seasonality effects [14,30]. Bipolar disorders' previously reported seasonality properties concerned patterns of individual symptoms, with manic episodes peaking during spring-summer and depressive episodes rising in the early winter

[33]. Similarly, previous schizophrenia-related studies had a different focus: They examined the risk associated with the season of a patient's birth, rather than the seasonality of disease relapse [31,34]. Schizophrenia's seasonality, as well as that of migraine, was hardly covered in the literature, yet many studies provide evidence for a connection between circadian rhythms and these 2 diseases [35–39]. Our results suggest that both conditions follow the characteristic annual cycle with summer recess, similar to other psychiatric conditions.

In contrast to our scarce understanding of psychiatric disorders' seasonality, annual prevalence variation is a well-established phenomenon in infectious disease epidemiology. Seasonal infection waves are driven by typically well-understood factors, such as seasonal transmission of infection, host behaviors, and seasonal variation of host susceptibility to infections [40,41]. Pronounced infectious disease seasonality aligns well with a priori expectation. In the present study, we used an analysis of infection seasonality as a positive control to corroborate the validity of both our method and our data. We still noticed some less obvious discordance. Verified in 2 nations, children smaller than 10 are less affected by ear infection in summer. Conversely, teenagers (11 to 20) are more likely to be infected in summer (Fig 4). This tendency continues in Sweden's older population in Sweden (21 to 50) and older males in the US (11 to 65). Another interesting observation is that UTI is seasonal only in some population groups: US males between 11 and 20 years old and Swedish females.

We observed a large difference in the magnitude of seasonality's annual variation between the US and Sweden—but only in psychiatric—not infectious—diseases. Diagnostic rate fluctuation is much greater in Sweden than in the US—for example, depression diagnoses rates plunge about 48% in Swedish females compared to 14% in U.S. citizen females during the summer. The difference in summer depression rates might be due to daylight exposure [8–11,42,43], as summer daylight extension (and daylight shortening in the winter) is much more extreme in proximal to the polar circle, as in Sweden's case, than in the continental US. Also, healthcare coverage and country-specific policies may explain curve differences. Without a universal healthcare system, such as the one in Sweden, people with chronic psychiatric conditions who lack comprehensive health insurance plans may be reluctant to seek medical care until their conditions becomes acute emergencies.

Besides, the observed seasonality in some psychiatric disorders may be rooted in certain culture-specific events. For example, in the summer, decreased ADHD visits in school-aged adolescent (Figs 3 and 6) may be associated with summer vacation travels, affecting both physicians and patients. Additionally, mental health awareness programs in some US K-12 schools may also increase psychiatric diagnoses during school time (spring and autumn) [44].

To conclude, it appears that psychiatric disorders follow a strong seasonal prevalence variation, closely resembling that previously described for unipolar depression. The most probable explanation for this observed seasonality, we believe, involves cyclic changes of exposure to solar light, which, in turn, affects circadian clock rhythms. In addition, this seasonality reflects a society's social rhythms, such as the patterns of summer vacations and certain mental health awareness programs in US K-12 schools [44]. The uncloaked, pervasive, and homogeneous seasonality encourages us to contemplate the influence of the sleep–wake cycle, light exposure, and circadian rhythms on the development of neuropsychiatric disorders and to be aware of mental health's seasonality and its implication on the healthcare system.

## Materials and methods

### Data and assumptions

The primary goal of this study was to model disease seasonality (and trends) in the US in recent years. To probe this question, we made use of 2 large, country-scale datasets, the IBM

Watson Health MarketScan dataset [18] containing the insurance claim records of over 150 million unique U.S. citizens and the Swedish National Health Register [19], detailing the health dynamics of virtually all Swedes—over 11 million unique people visible in the data. The US data cover the time interval between 2003 and 2014, while the Swedish data encompass an interval between 1980 and 2013.

The institutional review board (IRB) at the University of Chicago determined that the study is IRB exempt, given that patient data in both countries were preexisting and de-identified.

Although the database is one of the largest and most comprehensive collections of US insurance claims, it was built by merging asynchronous subsets collected by multiple private health insurers. As a result, the data are characterized by some properties that complicate our analysis (Fig 1). We identified at least 6 such issues and will briefly explain what they are, along with the efforts we made to address them in our analysis.

First, although MarketScan follows population health statistics for over a decade (2003 to 2014), most of the patients are "visible" to insurance records for a shorter time interval; patients were only enrolled in the insurance records for a few years, a few months, or, sometimes, even a few weeks, leading to "asynchronous enrollment." Due to heterogeneous insurance recruitment practices, groups of people simultaneously joining an insurer by no means resemble a random sample from the general US population. Second, possibly due to changes in coding standards, the dataset contains annual shifts (systematic jumps) of DRs at the beginning of every year for population strata of consistent composition. We observed these shifts for a subset of diseases. Third, some US holidays result in a general disruption of both health practice and reporting, and these disruptions are visible in the raw disease prevalence plots. Fourth, we noticed that most diseases manifest annual periodic prevalence fluctuations. For example, skin infections are on the rise in the summer, while upper respiratory system infections are more frequent in the winter. Fifth, disease prevalence seasonalities and trends vary greatly across sex and age-groups. Lastly, the data contain stochastic noise (temporal fluctuations in the recorded disease diagnoses) and, possibly, diagnosis encoding errors.

All these abovementioned factors influence the raw observations of diagnoses rates over time. One naive approach is to estimate the raw trend, treating the population as a whole, and fitting a line of DR, which we then calculate as the total number of diagnoses over the total number of enrollments across the whole database. This produces results somewhat discordant with the findings of previous published studies. Therefore, we designed a Bayesian multilevel model that addresses the issues discussed above (Fig 2) and which allowed us to infer latent disease trends for any specific age and sex group.

Additionally, we modeled the disease seasonalities (and trends) in Sweden based on their national registry, which incorporates almost all 9 million Swedes. Although there is no exact enrollment information supplied, we can safely assume static enrollment in years because Swedish patients were disenrolled only if they died or left Sweden. In other words, unlike the US dataset, the Swedish database is immune to the "asynchronous enrollment" problem.

## Methods and techniques

We modeled male and female trends separately. To make corrections for asynchronous patient enrollments, we first grouped all 150 million enrollees in the US database by (1) their enrollment dates (starts and ends); (2) patients' age at the middle of enrollment; and (3) patients' sex.

We then identified a collection of nearly a million enrollment range-, age-, and sex-specific population strata. Each stratum was characterized by a unique enrollment interval, for example, January 1, 2003 to December 31, 2004. We also placed different sexes into separate strata

and further subdivided patients by age-groups. Specifically, we used the following approximate, decade-long age subdivision: 0 to 10, 11 to 20, 21 to 30, 31 to 40, 41 to 50, and 51 to 65. Claims that occurred at over 65 years old were discarded because the majority of those enrollees supposedly switched to Medicare, and the remaining records for patients over 65 were likely to be erroneous or nonrepresentative.

For each population stratum, assuming a latent linear trend and a yearly repeated disease prevalence seasonality, we defined a model and estimated its parameters. However, this approach would become practically intractable if we were to fit this model on nearly 1 million population strata simultaneously. Therefore, we reduced the number of population strata by merging them into a smaller number of bigger strata, based on the proximity of enrollment boundaries, using $K$-means clustering. We pooled population strata with close enrollment boundaries together. In this way, we obtained around 600 "softbounded" population strata (100 for each age-group) for each sex. Each composite stratum is a combination of hundreds of raw, "hard-bounded" populations. These composite strata vary slightly in terms of date of enrollment beginning and end, typically in the range of a few weeks, and are rather homogeneous inside the shared enrollment window. Considering the Bayesian consistency of our model, a more robust and powerful way would be to cluster and merge population strata using a Bayesian Gaussian mixture model. However, such a method would soon exhaust a computer's random access memory because of the need to track a large number of variables. It is well known that the $K$-means method is equivalent to the Gaussian mixture model's hard EM (expectation–maximization) implementation in some limiting formulation. Here, we argue that $K$-means clustering, chosen for its scalability, is good enough to mimic the behavior of the Bayesian Gaussian mixture model and accomplish population number reduction.

For each disease, we decomposed the DR trend, for a given softbound population stratum $i$ of age-group $j$ for a sex-specific condition, into 4 parts: a linear trend, possible shifts at the beginning of every year, a seasonality term modeling periodic patterns, and an error term incorporating all other effects.

$$y_{i,j}(t) = \text{Linear trend} + \text{Yearly shifts} + \text{Seasonality} + \text{Error}, \tag{1}$$

where

$$\text{Linear trend} = \alpha_{i,j} + \beta_{i,j}t, \tag{2}$$

$$\text{Yearly shifts} = \sum_{k} \mathbf{1}(t > s_k) \cdot \gamma_{i,j,k}, \tag{3}$$

$$\text{Seasonality} = \sum_{n=1}^{N} p_{i,j,n}\cos\frac{2n\pi t}{W} + \sum_{n=1}^{N} q_{i,j,n}\sin\frac{2n\pi t}{W}, \tag{4}$$

$$\text{Error} = \epsilon_{i,j}. \tag{5}$$

In the above equations, $y_{i,j}(t)$ is the DR of a softbound population stratum $i$ of age-group $j$ at time point (week) $t$. Parameters $\alpha_{i,j}$ and $\beta_{i,j}$ are the intercept and the slope, respectively, of the latent linear trend. Moreover, $\mathbf{1}$(condition) is an indicator function that evaluates to 1 only if the input condition is true. $s_k$ and $\gamma_{i,j,k}$ are the $k$th separation and shift, respectively. The separations are when the shift could happen, and we assumed they are all year starts ($s_1$ = January 1, 2003, $s_2$ = January 1, 2004 . . .).

Note that for the Swedish database, all enrollees are visible from the start, so there is no "asynchronous enrollment" problem and only 1 all-inclusive population stratum was considered for each age-group and sex.

We used a Fourier series with period $W = 365.25/7$ weeks to model the potential seasonality of some conditions. $p_{i,j,n}$ and $q_{i,j,n}$ are harmonic bases.

The traditional parametrization of the "seasonality term" (a Fourier series) is convenient for the estimation phase of analysis. However, to interpret estimates, it is intuitive to use the following re-parametrization of the Fourier term:

$$\text{Seasonality} = \sum_{n=1}^{N} A_{i,j,n} \sin\left(\frac{2n\pi t}{W} + \phi_{i,j,n}\right), \tag{6}$$

where

$$A_{i,j,n} = \sqrt{p_{i,j,n}{}^2 + q_{i,j,n}{}^2}, \text{ and} \tag{7}$$

$$\phi_{i,j,n} = \text{Arctan2}(p_{i,j,n}, q_{i,j,n}). \tag{8}$$

$A_{i,j,n}$ and $\phi_{i,j,n}$ in Eq 6 are amplitude and phases for the $n$'s harmonic. Arctan2 corresponds to a 2-argument arctangent function.

We estimated all parameters under a Bayesian framework. We sampled the prior values of $\alpha_{i,j}$ and $\beta_{i,j}$ from skew normal distributions with age-group–specific locations, scales, and shapes. A skew normal distribution density function is defined in the following way:

$$f(x; \text{loc} = \mu, \text{ scale} = \sigma, \text{ shape} = h) = \frac{2}{\sigma}\phi\left(\frac{x-\mu}{\sigma}\right)\Phi\left(h\left(\frac{x-\mu}{\sigma}\right)\right), \tag{9}$$

where $\phi(x)$ and $\Phi(x)$ are density and cumulative distribution functions, respectively, for a standard normal distribution. Our choice of prior distribution was motivated by an analysis of the parameter estimate distributions for various groups of patients—they indeed resemble the skew normal shape.

$$\alpha_{i,j} \sim \text{SkewNormal}(\text{loc} = \mu_j^\alpha, \text{ scale} = \sigma_j^\alpha, \text{ shape} = h_j^\alpha), \tag{10}$$

$$\beta_{i,j} \sim \text{SkewNormal}(\text{loc} = \mu_j^\beta, \text{ scale} = \sigma_j^\beta, \text{ shape} = h_j^\beta). \tag{11}$$

To allow information flow through different age-groups, we sampled the location parameters from a zero-mean Gaussian process prior:

$$\mu_j^\alpha = \mu^\alpha(j) \sim \text{GaussianProcess}(0, k^\alpha(j, j')), \tag{12}$$

$$\mu_j^\beta = \mu^\beta(j) \sim \text{GaussianProcess}(0, k^\beta(j, j')), \tag{13}$$

where $k^\alpha(j, j')$ and $k^\beta(j, j')$ are exponentiated quadratic kernels with scale and length drawn from flat hyperpriors. The prior of the Gaussian process "linked" different age-groups within a unified estimation procedure and allowed information about disease trend flow across age-groups—by assuming that similar age-groups share similar trend parameters.

We drew the scale parameters of $\alpha_{i,j}$ and $\beta_{i,j}$ from flat, half-Cauchy hyperpriors, and we restricted the shape parameters $h_j^\alpha$ and $h_j^\beta$ by zero-mean Laplace distributions so that the scale parameter would not compete with the shape parameter. In our experiments, we found that a

skew normal with a large shape could behave similarly as a skew normal with a large scale. This pathological behavior would result in inefficient sampling.

We sampled the population as well as age-specific shifts $\gamma_{i,j,k}$ from a zero-mean Laplace distribution, thus incorporating our prior belief that shifts should not mask the linear trend effect. We sampled the bases for seasonality from zero-mean normal distributions.

Finally, to offset the effect of holidays and celebrations, we applied a holiday-smooth function that took the average DRs around US federal holidays and Easters/Good Fridays. We overcame the presence of outliers caused by other unknown forces using a Student $t$ distribution sampling:

$$\text{HolidaySmooth}[y_{i,j}(t)] \sim \text{StudentT}(\mu_{i,j}^{y}, \sigma_{i,j}^{y}, v_{i,j}^{y}), \tag{14}$$

where the location parameter to $\mu_{i,j}^{y}$ = Linear trend + Yearly shifts + Seasonality. $\sigma_{i,j}^{y}$ and $v_{i,j}^{y}$ are scales and degrees of freedoms sampled from flat half-Cauchy hyperpriors. Fig 1B illustrates the outcome of the holiday-smooth function.

We approximated the model using a No-U-Turn sampler [20] initialized by variational inference [45]. In general, for one sex-specific condition, it would take hundreds of CPU hours to attain a reasonable estimation due to the high-dimensional searching space—we were sampling thousands of parameters simultaneously. We applied the model to 33 neuropsychiatric and 47 infectious conditions of 2 sexes and tried to reproduce and make corrections for trends in different age-groups.

Once we obtained the estimation of harmonic bases $p_{i,j,n}$ and $q_{i,j,n}$ as in Expression (4), we computed the posterior harmonic base estimates for the whole population as

$$\bar{p}_{j,n} = \sum_{i} w_i \cdot p_{i,j,n}, \tag{15}$$

$$\bar{q}_{j,n} = \sum_{i} w_i \cdot q_{i,j,n}, \tag{16}$$

where $w_i$ is the weight according to the size of population stratum $i$, so $\bar{p}_{j,n}$ and $\bar{q}_{j,n}$ could be interpreted as the estimate of $n$th harmonic bases for age-group $j$ for the whole population.

After Bayesian inference, we obtained all the posterior parameter distributions, which enabled us to estimate the annual seasonality free from the influence of trends, sudden shifts, and noises such as holiday effects. For each age-/sex-specific condition, we divided the raw DR seasonality to its time-average DR over 570 weeks (starting from January 1, 2003, Expressions (17) and (18)) and obtained the relative fluctuation in percentage, as shown in the main figures.

$$\text{time}-\text{average DR} = \langle \text{DR} \rangle = \frac{1}{570} \sum_{w=1}^{570} \text{DR}_{w} \tag{17}$$

$$s(t) = \frac{\text{Seasonality estimate}}{\langle \text{DR} \rangle} \tag{18}$$

We attempted to correct baseline fluctuation of all medical visits by deducting $s(t)$ from the $s_{\text{all}}(t)$ that represents the yearly variation of all conditions and diseases (S1 Fig and Fig 2, Step 2 lower center plot):

$$\text{Corrected } s'(t) = s(t) - s_{\text{all}}(t) \tag{19}$$

This procedure also revealed disease trends. However, as we carefully examined these estimates, it was clear that they might not reflect real disease trends over time simply because we estimated the trends with cohorts of quasi-static enrollments—the same people joined and left

and their age went up accordingly. The change of age drastically impacted our disease trend estimations. For example, we observed that incidents of some infectious diseases went down because the prevalence of some pediatric infections decreases as children grow older. By contrast, many cardiovascular condition trends are positive because older people are more prone to them.

Fig 2 summarizes our model, where data corresponding to a "raw" trend are an input to our model. A "raw" trend is deconvoluted into trends within hundreds of population strata based on enrollment dates (the left panel and the center panel). The model fits each population stratum separately, but still allows certain information shared across population strata, in a hierarchical framework (the center panel). Finally, we make corrections and estimate the seasonalities and trends for specific age-groups (the right panel).

Lastly, it is worth mentioning that we dropped all higher-order harmonics in the Fourier series after the first 5 ($N = 5$) for approximation based on model selection results (Eqs 4 and 6). We tested $N = 5$, 15, and 25 to find the best approximation model. To evaluate the model, we computed the sum of Watanabe–Akaike information criteria (WAIC) [46] over 33 neuropsychiatric and 47 infectious diseases in the 2 sexes and found that $N = 5$ was simple and good enough to model the seasonality (S13 Fig).

The Bayesian procedure we designed helped to mitigate multiple confounding factors with a multilevel model, but it could also be problematic, given its complexity. First, we could not certify the convergence of the MCMC because we were not estimating one single parameter, whereby a diagnostic statistic like Gelman–Rubin [47] or Geweke [48] would have been applicable to determine the mixing of that parameter. The approximation of each disease's seasonality involved thousands of parameters, making it difficult to determine how many iterations were needed to reach a stationary point. To alleviate this concern, we employed the No-U-Turn sampler, which is able to mix an MCMC process rapidly and reliably [20]. More importantly, we inspected the disease trend's posterior expectation curves and seasonality, restored from the posterior estimation of parameters (like the green lines on Fig 2, Step 2, upper panel) and confirmed that they were aligned with the input raw trend and seasonality. Collectively, using all the available tools, the intrinsic seasonality is reflected in the results insofar as we are able.

## Supporting information

**S1 Data. All result plots for 4 high-latitude states in the US (AK, WA, MT, ND, or AWMN), the whole US, 2 low-latitude states (TX and FL), and SE, split file part 1.** For each regional analysis, we supply the DR trend's posterior estimation, compared to the raw observational trend to show how well the Bayesian model fits the data. We also provide the corrected and uncorrected seasonality plots for all sex–age groups for each tested disease. AK, Alaska; AWMN, Alaska, Washington, Montana, North Dakota; DR, diagnosis rate; FL, Florida; MT, Montana; ND, North Dakota; SE, Sweden; TX, Texas; WA, Washington.
(001)

**S2 Data. All result plots for 4 high-latitude states in the US (AK, WA, MT, ND, or AWMN), the whole US, 2 low-latitude states (TX and FL), and SE, split file part 2. AK, Alaska; AWMN, xxx; FL, Florida; MT, Montana; ND, North Dakota; SE, Sweden; TX, Texas; WA, Washington.**
(002)

**S3 Data. All result plots for 4 high-latitude states in the US (AK, WA, MT, ND, or AWMN), the whole US, 2 low-latitude states (TX and FL), and SE, split file part 3. AK,**

Alaska; AWMN, xxx; FL, Florida; MT, Montana; ND, North Dakota; SE, Sweden; TX, Texas; WA, Washington.

(003)

**S4 Data. All result plots for 4 high-latitude states in the US (AK, WA, MT, ND, or AWMN), the whole US, 2 low-latitude states (TX and FL), and SE, split file part 4. AK, Alaska; AWMN, xxx; FL, Florida; MT, Montana; ND, North Dakota; SE, Sweden; TX, Texas; WA, Washington.**

(004)

**S5 Data. All result plots for 4 high-latitude states in the US (AK, WA, MT, ND, or AWMN), the whole US, 2 low-latitude states (TX and FL), and SE, split file part 5. AK, Alaska; AWMN, xxx; FL, Florida; MT, Montana; ND, North Dakota; SE, Sweden; TX, Texas; WA, Washington.**

(005)

**S6 Data. All result plots for 4 high-latitude states in the US (AK, WA, MT, ND, or AWMN), the whole US, 2 low-latitude states (TX and FL), and SE, split file part 6. AK, Alaska; AWMN, xxx; FL, Florida; MT, Montana; ND, North Dakota; SE, Sweden; TX, Texas; WA, Washington.**

(006)

**S7 Data. All result plots for 4 high-latitude states in the US (AK, WA, MT, ND, or AWMN), the whole US, 2 low-latitude states (TX and FL), and SE, split file part 7. AK, Alaska; AWMN, xxx; FL, Florida; MT, Montana; ND, North Dakota; SE, Sweden; TX, Texas; WA, Washington.**

(007)

**S8 Data. All result plots for 4 high-latitude states in the US (AK, WA, MT, ND, or AWMN), the whole US, 2 low-latitude states (TX and FL), and SE, split file part 8. AK, Alaska; AWMN, xxx; FL, Florida; MT, Montana; ND, North Dakota; SE, Sweden; TX, Texas; WA, Washington.**

(008)

**S9 Data. All result plots for 4 high-latitude states in the US (AK, WA, MT, ND, or AWMN), the whole US, 2 low-latitude states (TX and FL), and SE, split file part 9. AK, Alaska; AWMN, xxx; FL, Florida; MT, Montana; ND, North Dakota; SE, Sweden; TX, Texas; WA, Washington.**

(009)

**S10 Data. All result plots for 4 high-latitude states in the US (AK, WA, MT, ND, or AWMN), the whole US, 2 low-latitude states (TX and FL), and SE, split file part 10. AK, Alaska; AWMN, xxx; FL, Florida; MT, Montana; ND, North Dakota; SE, Sweden; TX, Texas; WA, Washington.**

(010)

**S1 Fig. The baseline seasonality of all medical visits in the 4 high-latitude states in the US (AK, WA, MT, ND, or AWMN), the whole US, 2 low-latitude states (TX and FL), and SE.** The data underlying this figure can be found in https://doi.org/10.5061/dryad.vdncjsxv6. AK, Alaska; AWMN, xxx; FL, Florida; MT, Montana; ND, North Dakota; SE, Sweden; TX, Texas; WA, Washington.

(TIF)

**S2 Fig. The uncorrected seasonality of skin infection in the US.** The data underlying this figure can be found in https://doi.org/10.5061/dryad.vdncjsxv6.
(TIF)

**S3 Fig. The uncorrected seasonality of psychiatric diseases in the 4 high-latitude states: AK, WA, MT, and ND.** The data underlying this figure can be found in https://doi.org/10.5061/dryad.vdncjsxv6. AK, Alaska; MT, Montana; ND, North Dakota; WA, Washington.
(TIF)

**S4 Fig. The uncorrected seasonality of infectious diseases in the 4 high-latitude states: AK, WA, MT, and ND.** The data underlying this figure can be found in https://doi.org/10.5061/dryad.vdncjsxv6. AK, Alaska; MT, Montana; ND, North Dakota; WA, Washington.
(TIF)

**S5 Fig. The uncorrected seasonality of psychiatric diseases in the 2 low-latitude states: TX and FL.** The data underlying this figure can be found in https://doi.org/10.5061/dryad.vdncjsxv6. FL, Florida; TX, Texas.
(TIF)

**S6 Fig. The uncorrected seasonality of infectious diseases in the 2 low-latitude states: TX and FL.** The data underlying this figure can be found in https://doi.org/10.5061/dryad.vdncjsxv6. FL, Florida; TX, Texas.
(TIF)

**S7 Fig. The uncorrected seasonality of schizophrenia-related psychosis in the US and SE.** The data underlying this figure can be found in https://doi.org/10.5061/dryad.vdncjsxv6. SE, Sweden.
(TIF)

**S8 Fig. The uncorrected seasonality of migraine in the US and SE.** The data underlying this figure can be found in https://doi.org/10.5061/dryad.vdncjsxv6. SE, Sweden.
(TIF)

**S9 Fig. The corrected seasonality of psychiatric diseases in the 4 high-latitude states: AK, WA, MT, and ND.** The data underlying this figure can be found in https://doi.org/10.5061/dryad.vdncjsxv6. AK, Alaska; MT, Montana; ND, North Dakota; WA, Washington.
(TIF)

**S10 Fig. The corrected seasonality of infectious diseases in the 4 high-latitude states: AK, WA, MT, and ND.** The data underlying this figure can be found in https://doi.org/10.5061/dryad.vdncjsxv6. AK, Alaska; MT, Montana; ND, North Dakota; WA, Washington.
(TIF)

**S11 Fig. The corrected seasonality of psychiatric diseases in the 2 low-latitude states: TX and FL.** The data underlying this figure can be found in https://doi.org/10.5061/dryad.vdncjsxv6. FL, Florida; TX, Texas.
(TIF)

**S12 Fig. The corrected seasonality of infectious diseases in the 2 low-latitude states: TX and FL.** The data underlying this figure can be found in https://doi.org/10.5061/dryad.vdncjsxv6. FL, Florida; TX, Texas.
(TIF)

**S13 Fig. The model selection for choosing the number of harmonics.** The model with $N = 5$ has the lowest sum of WAIC over 33 psychiatric and 47 infectious diseases. It suggests the simpler model is good enough to model disease seasonality. In the example of depression in young males, adding up harmonics would not help the estimation, given the intrinsic simplicity of seasonality. The data underlying this figure can be found in https://doi.org/10.5061/dryad.vdncjsxv6. WAIC, Watanabe–Akaike information criteria.
(TIF)

## Acknowledgments

We are grateful to E. Gannon, R. Melamed, and M. Rzhetsky for comments on earlier versions of this manuscript.

## Author Contributions

**Conceptualization:** Hanxin Zhang, Henrik Larsson, Andrey Rzhetsky.

**Data curation:** Hanxin Zhang, Atif Khan, Qi Chen, Henrik Larsson.

**Formal analysis:** Hanxin Zhang, Atif Khan, Qi Chen, Andrey Rzhetsky.

**Funding acquisition:** Andrey Rzhetsky.

**Investigation:** Hanxin Zhang, Qi Chen, Andrey Rzhetsky.

**Methodology:** Hanxin Zhang, Atif Khan, Qi Chen, Andrey Rzhetsky.

**Project administration:** Henrik Larsson, Andrey Rzhetsky.

**Software:** Hanxin Zhang.

**Supervision:** Andrey Rzhetsky.

**Validation:** Hanxin Zhang.

**Visualization:** Hanxin Zhang.

**Writing – original draft:** Hanxin Zhang, Andrey Rzhetsky.

**Writing – review & editing:** Hanxin Zhang, Atif Khan, Qi Chen, Henrik Larsson, Andrey Rzhetsky.

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
