## [Editor Report · Decision Letter 0]

24 Sep 2020

Dear Andrey, 

Thank you for submitting your manuscript entitled "Are Psychiatric Disorders as Seasonal as Infections?" for consideration as a Short Report by PLOS Biology.

Your manuscript has now been evaluated by the PLOS Biology editorial staff, as well as by an academic editor with relevant expertise, and I'm writing to let you know that we would like to send your submission out for external peer review.

IMPORTANT: After some discussion with the Academic Editor and the team, we think this article might be better considered as a Discovery Report (https://journals.plos.org/plosbiology/s/what-we-publish#loc-linked-articles). The reason for this is that the Academic Editor felt that greater insights would be provided into the phenomena that you describe if you were to perform a subsequent study with a cohort from the Southern Hemisphere. Such a follow-up study could be submitted to PLOS Biology (by you or by another group) as an Update Article, and linked to your Discovery Report. It might also be helpful to pre-register this subsequent study. At the moment no formatting changes are needed, but can you change the article type to "Discovery Report" when you upload the additional metadata required (see next paragraph)?

Please re-submit your manuscript within two working days, i.e. by Sep 28 2020 11:59PM.

Kind regards,

Roli

Senior Editor

PLOS Biology

---

## [Decision Letter · Decision Letter 1]

31 Dec 2020

Dear Andrey,

Thank you very much for submitting your manuscript "Are Psychiatric Disorders as Seasonal as Infections?" for consideration as a Research Article at PLOS Biology. Your manuscript has been evaluated by the PLOS Biology editors, an Academic Editor with relevant expertise, and by three independent reviewers. Please accept my further apologies for the time it has taken to obtain appropriate advice during these challenging circumstances.

You'll see that while the reviewers are intrigued, they each raise a number of concerns that will need to be addressed for further consideration. Some of these, including those from reviewer #1, who is familiar with the Swedish dataset, seem potentially problematical. The concerns include the exclusion of potential confounds, the need to use more appropriate comparators, and requests for further analyses and presentational changes.

In light of the reviews (below), we will not be able to accept the current version of the manuscript, but we would welcome re-submission of a much-revised version that takes into account the reviewers' comments. We cannot make any decision about publication until we have seen the revised manuscript and your response to the reviewers' comments. Your revised manuscript is also likely to be sent for further evaluation by the reviewers.

We expect to receive your revised manuscript within 3 months. 

**IMPORTANT - SUBMITTING YOUR REVISION**

*Re-submission Checklist*

*Published Peer Review*

*PLOS Data Policy*

*Blot and Gel Data Policy*

Best wishes and Happy New Year,

Roli

Senior Editor,

rroberts@plos.org,

PLOS Biology

REVIEWERS' COMMENTS:

Reviewer #1:

This paper uses two large healthcare datasets to examine temporal fluctuation in major psychiatric conditions in the USA and Sweden. Despite the impressive datasets and statistical analyses performed there are a number of critical conceptual problems with the paper, as follows:

1. The hypothesis and title are poorly articulated. The research question in the title is not directly answered in the paper, though does present evidence that the cyclical patterns of psychiatric disorders are very different to infectious diseases. If I interpret their results correctly, psychiatric disorders (or hospitalised diagnoses therefor) are more patterned than infectious diseases. The hypothesis underlying the article is further based on conjecture that any patterns are due to the "annual light-dependency cycle" but this cannot and is not tested in the paper (the paper lacks data on individual or even ecological exposure to sunlight hours (which would also vary by latitude in both Sweden and the USA)) or, moreover, the circadian cycles on which the discussion speculates. The patterns may be driven by a variety of other causal effects - most of which are equally untestable - the role of vitamin D, for example, or changes in exercise levels. As a result, I feel the paper reaches in places. 

2. I am not an expert on the US data, but have worked with Swedish register data on a regular basis. The paper does not provide sufficient detail on how the diagnostic data and cohort was determined from the National Patient Register. For example, did the study only consider the primary diagnosis at each visit, or any of up to 20 (or more) diagnostic codes which may be made at a single visit. Did the analyses take into account the clustering of repeated diagnoses ("relapses") within individuals, and how did the authors distinguish between spells of care for the same episode, from a relapsed episode? I am unaware of any definition of remission which is possible from using the NPR, but the paper mentions they were interested in remission, so I am curious how this was operationalised in Sweden (and for that matter in the US data - was it simply the absence of diagnosis?). 

3. The NPR in Sweden is not suitable for studying depression, as most cases of depression do not get treated in secondary care, but in primary care, not covered by the register. This seems like the generalisability of the findings may be limited to the severest cases. 

4. The apparent patterns in Sweden may be entirely attributable to artefact of the Swedish holiday system where during the summer months, a substantial proportion of the population go on vacation to summer homes and so forth (between June-August). This seems entirely possible to account for the lower levels of diagnoses made during this period, and likewise the dip in the winter months for several psychiatric conditions around Christmas and New Year, which would seem to further argue against a circadian hypothesis. The peak months for most psychiatric conditions are spring and autumn, presumably coinciding with periods when health services are operating closer to their nominal workforce capacity.

5. Seasonal effects for schizophrenia have been consistently demonstrated, but these are associated with a Winter-Spring birth and effects on neurodevelopment, not the factors immediately precipitating onset/remission. It feels like a missed opportunity to have delved deeper into this issue. 

6. Many elements of the methodology appear in the results section and this interrupts the presentation of the data. 

7. In the abstract I am unclear what "seasonal affective depression" means - the authors seem to have conflated two diagnoses. 

Reviewer #2:

The main problem with the paper is the inability to properly rule out illness behaviour as the main driver of the seasonal effects. This is difficult to do, however one specific strategy I would implement is to replace infectious diseases as the comparator, instead using miscellaneous chronic medical illnesses instead. Acute infectious diseases are the worst possible comparator for this purpose because they are the least likely disorders to be driven by illness behaviour..rather they are almost 100% driven by seasonal biological realities. Chronic medical conditions on the other hand would offer an excellent control for general illness behaviour and other non-specific effects that might have a seasonal basis.

Further to this point, more consideration of overall seasonal patterns across all psychiatric and medical diagnoses would be of interest....for example I wondered whether the data could be presented as ratios, using total medical visits as the denominator. Truly seasonal disorders should have a greater proportional prevalence relative to all disorders across seasons....using total visits as a control would help in this way.

The argument that holiday patterns are different across locations is not strong....many people change their lifestyle and priorities when the weather is nice whether or not on vacation. 

The authors should mention that school/work pressure in the fall may explain the ADHD seasonal effect. They should also discuss a now solid group of studies showing that ADHD patients have high rates of seasonal depression and phase delays in sleep rhythms.

Reviewer #3:

[identifies herself as Micaela E. Martinez]

Dear Authors and Editors,

I enjoyed reviewing the submitted manuscript "Are Psychiatric Disorders as Seasonal as Infections?" by Zhang et al. In this manuscript, the authors use two extensive datasets to test for seasonality in psychiatric diseases. The data were quite striking and the results clearly demonstrated seasonality among psychiatric diseases. The analysis was relatively straightforward and their results clearly communicated. The conclusion that psychiatric diseases (specifically ADHD, substance abuse, adjustment disorder, anxiety phobic disorder, and depression) are elevated in the spring, autumn, and winter has great public health importance. I have provided comments below for additional detail and discussion that I believe would be needed before publication. 

Overall, this study is novel with significant findings and has an appropriate statistical design. It is not clear to me if the data and example code needed to replicate the analysis are provided. The authors should indicate if their aggregated time series will be placed on Dryad or another digital repository. 

Sincerely,

Micaela E. Martinez, Columbia University

(I provide non-anonymous review in support of double-open or double-blind review protocols)

Major comments.

It would be helpful to explain why these five particular psychiatric diseases were chosen. Were these the most prevalent in your datasets?

It would be helpful to see the seasonal curves for the five diseases on Fig 1 all on a single plot in order to see the similarities in the seasonality. 

It would be helpful to see the raw data and posterior curves for each psychiatric disease with data from all age groups and both sexes aggregated (i.e., similar to the raw observations shown in Fig S8 panel D left and the posterior expectation on the right). 

For the psychiatric diseases, it would be helpful to see the seasonality on a polar coordinate where the summer solstice, winter solstice, autumn equinox, and spring equinox are marked so we may clearly see the seasonal pattern relative to photoperiod extremes and transitions.

It would be helpful to explore the diagnosis rate relative to daylength in each country. The seasonal change in photoperiod is much more extreme in Sweden than the US (and this could be an important feature, as discussed by the authors) and a formal analysis of the daylength effect could be done for the supporting information. 

line 79. It reads as though the data will be analyzed in 4 seasonal bins, whereas, the seasonal curves are actually in a daily or weekly resolution (from what I can infer). It is important to note that the 4 seasons are being used as reference points for discussion and not for aggregating the data into seasonal bins. It will also be important to note if the time series were daily or weekly. I am assuming daily based on the holiday effect mentioned in Fig S8.

line 105. Following on the comment above, are the curves in Fig 1 derived from daily data? Because above you only mention the 4 seasonal windows for the data. The methods suggest daily or weekly resolution.

Fig 1. Again, adding to the comment above, since the results come before the discussion, please describe how the curves are derived. I am assuming the analysis is using weekly or daily data because there is such a great discrepancy between min and max values for some curves. For instance, the max is 1/2 the min for depression in Sweden for 11-20 yr olds. 

Fig 1. It would be interesting to include a diagnosis that doesn't have a biological basis as a "control". Something like injuries from accidents - we wouldn't expect to be seasonal. Otherwise people may argue that the observed seasonal patterns are due to less healthcare seeking behavior in summer. 

Fig S3. For all of the figures showing the seasonality, such as Fig 1 and Fig S3, it would be helpful to have a y-axis scale to tell which age group has the highest incidence. For instance, in Fig S3 is the seasonality of ear infection simply more detectable in the youngest age group because this is the group with the highest incidence of ear infection? If there were a y-axis scale, the reader would be able to compare the incidence across age groups. 

line 231. As mentioned above, it would be good to have a 'negative control', something you don't expect to be seasonal because it doesn't have a biological basis, such as injury from accidents.

lines 233-239 . As mentioned above, it would be important to also address the difference in incidence among age groups. For example, in Sweden, the trough in ear infection in 0-10 is in the summer. I would expect that a large fraction of ear infection occurs in the 0-10 yr old group; thus, is the seasonality in this group the most "representative" seasonality for ear infection? It might be that the ability to detect the seasonality in some age groups is limited due to low incidence and therefore it is important to show the actual number of cases in each group, or cases per 100,000. Following on my comment above, it would be good to show the curve for each disease where males and females and all age groups are aggregated. 

Discussion section. It is necessary in the discussion section for the authors to discuss how circadian rhythms relate to seasonality, because it will be confusing for many readers. Perhaps the authors can reference more of the literature on SAD and also the literature on melatonin seasonality (e.g., papers by Thomas Wehr and by Ken Wright). 

Discussion section. It would also be helpful for the authors to provide some biological information about the psychiatric disorders in this study so readers can have a better understanding of how clocks could play a role. Are these disorders impacted by hormones, the immune system, sleep, or metabolism? Also, it could be good for the authors to discuss seasonal light exposure in general and how this may impact circadian rhythms. 

line 435. This paragraph needs more explanation. The way it reads to me is that the parameters for the trend, shifts, and noise terms were estimated and then the seasonal remainder was estimated last. Is this correct? Or were all parameters estimated simultaneously? Or estimated in a specific order? When it says the authors "divided the raw DR seasonality to its average DR over 570 weeks", does this mean they took the seasonal component from [Disp-formula pbio.3001347.e010] and divided it by the linear trend + yearly shift from [Disp-formula pbio.3001347.e010]? Or that they took the seasonal component and divided it by the mean of the yearly trend + yearly shift across all years?

lines 441. As mentioned above, due to the age group trends, it would be important to calculate a prevalence/incidence estimate. For example, calculate diagnosis per 100,000 patients in each age group. The seasonal curves that are most important from a public health perspective are those most representative of the disease. Thus, the childhood curves will be most important for the pediatric diseases and the older age groups will be most important for the diseases that usually manifest in adulthood. 

lines 450. Please provide what the four center panels in figure S8D are showing, which age groups are shown here? Can you also be more explicit about which parameters are shared among the age groups (within a disease)? You say that "The model fits each population in a hierarchical way so that information is shared across populations", but it seems as though each age group has their own trend and seasonality, so what information has been shared/constrained in this hierarchical fitting?

Fig S8. It would be helpful to have panels A, B, and D in the main text of the manuscript to give readers a feel for the modeling workflow. Also, panel A needs a better description in the caption. For example, it is not clear what is meant by "naive fitting line". Also, in this panel, shouldn't the "periodic patterns" curve not have a trend in it? I believe it should be flat unless it is illustrating the periodic + trend.

Minor comments.

line 20. "We found that psychiatric diseases' annual patterns are remarkably similar across the studied diseases in both countries, with the magnitude of annual variation significantly higher in Sweden than in the US for psychiatric, but not infectious diseases, potentially pinning the pathogenesis of psychiatric diseases on circadian rhythms." 

I suggest rephrasing the sentence above. The fact that (for infectious diseases) the magnitude of seasonality doesn't vary between countries does not tell us anything about clock involvement or lack thereof. This is because acute infectious disease seasonality is shaped by the transmission process. I would recommend breaking the above quote into two sentences. Such as: We found that psychiatric diseases' annual patterns are remarkably similar across the studied diseases in both countries, with the magnitude of annual variation significantly higher in Sweden than in the US for psychiatric, but not infectious diseases. The seasonality of psychiatric diseases suggests the pathogenesis of psychiatric diseases may be driven circadian rhythms…[then explain the logic of how circadian rhythms relate to seasonality]

line 28. The first sentence sounds a bit off-putting because it sounds like the authors are suggesting that psychiatric disorders are mental constructs rather than biological diseases. This could be offensive to some readers.

line 111. I am confused about this web link. Are these the manuscript data on this website? Were the analyses on this website conducted by another research group? If this is the manuscript data, I suggest it be included and described in supplemental info.

line 227. The examples "seasonal access to pathogens" and "summer swimming activities" are a bit misleading. It would be better to mention seasonal transmission of infection and host behavior.

lines 237. This sentence is problematically speculative, especially since the authors don't show that ear infection is caused by outdoor activity. I suggest removing this sentence: 'It is then plausible to impute this dissimilarity between age groups and sexes to different levels of outdoor activities in the summer, which entails, speculatively, that American males are more active in summer than American females whereas the gender gap is minimal in Sweden.' 

line 262-300. I am not sure of the current formatting requirements for plos bio, but the "assumptions, material and methods section", if it will be in the main text, could be reformatted to "materials and methods" with the list of 6 items presented in more of a paragraph style. This would make the materials and methods section read more seamlessly with the other main text sections. 

lines 282. I recommend removing this example, it reads as ageist and may perpetuate stereotypes "one group can correspond to mostly healthy young factory workers, while another one could comprise aging oil refinery workers."

Discussion. Following on my comments above, it would be helpful in the discussion and in Fig 1 to mention that the seasonality was modeled with daily or weekly resolution. 

line 374. In Eq 7 and 8, please provide more detail of why this specific formulation is being used for the phase and amplitude. Is there a specific constraint enforced by using the p_{i,j,n} and q_{i,j,n} formulations?

Fig S8. Panel D, right panel - it would be helpful for readers if you indicate the shift happening in 2009, and perhaps show the seasonality curve, instead of just the age trends and the posterior.

---

## [Decision Letter · Decision Letter 2]

14 Jun 2021

Dear Andrey,

Thank you for submitting your revised Discovery Report entitled "Probing annual disease incidence cycles in US and Sweden" for publication in PLOS Biology. I have now obtained advice from the original reviewers and have discussed their comments with the Academic Editor. Many thanks for your patience while we sought this additional input; as you can imagine, a pandemic is an especially busy time for those reviewers who are epidemiologists.

Based on the reviews, we will probably accept this manuscript for publication, provided you satisfactorily address the remaining points raised by the reviewers. Please also make sure to address the following data and other policy-related requests.

IMPORTANT:

a) You'll see a diversity of opinion among the reviewers here. Reviewer #2 is largely satisfied, but questions whether the suggested approach was the correct one. We believe that because this approach was also suggested by reviewer #3, who is very well-placed to judge, and as she is now broadly happy with the outcome, that it was indeed a valid way forward...

b) ...however, because of your decision to also present the uncorrected analyses (which we will respect), this has resulted in some confusion, especially because of the somewhat strong language that you use to dismiss the corrected version (e.g. "nonsensical" is a term highlighted by both reviewers #2 and #3). You will see that reviewer #1, who recommended that we now reject your paper, finds this problem to be fatal....

c) ...to my thinking, the key thing to bear in mind when revising your paper to address these disparate remarks is to make the paper as clear as possible for the reader, while avoiding misleading them. We strongly recommend that you present both sets of results even-handedly, and with full transparency, until the Discussion, when you should present your arguments as to why you believe that the uncorrected analysis better captures the real picture. This would enable readers to "make up their own minds" while allowing you to present your case. We assume that a solid "answer" must await further research.

d) So: please revise your text with my comments (above) in mind, and asking yourself what reviewer #1 might think. For example, in the Abstract, instead of "Comparing two sets of results in context of published psychiatric disease seasonality studies, we tend to believe that our uncorrected results are likely to capture the real trends, while the corrected results reflect mostly artefacts generated by idiosyncratically fluctuating volumes of patient health-seeking visits across year" I would suggest something like "In the context of published psychiatric disease seasonality studies, we discuss whether our uncorrected results or the corrected ones are more likely to capture the real underlying trends."

e) Because we do not feel that you can reach a solid conclusion, we think that it would be better to keep an interrogative title (perhaps something like "Does psychiatric disease follow annual disease cycles?" which I note is quite similar to your original title). As US and Sweden are namechecked clearly in the Abstract, we don't think you need them in the title.

f) Please attend to all the remaining requests from reviewer #3.

g) Because of the extra analyses, we can no longer consider this paper as a Discovery Report; this is now a much more substantial analysis, and should be published as a full Research Article. Please could you change the article type to "Research Article" when you re-submit?

h) Please could you update your blurb to reflect the revision and the above comments?

i) Please could you attend to my Data Policy requests below. I note that your raw data are third-party, and I have treated these as for your previous PLOS Biology paper, which I understand used the same datasets. However, we will need the numerical values that are shown in the Figures; the location of these data should be cited clearly in all of the relevant legends.

j) I note that in your previous PLOS Biology paper you included a statement clarifying the ethics situation, namely "The University of Chicago IRB determined that the study is IRB exempt, given that patient data in both countries were pre-existing and de-identified." If correct, could you possibly include an equivalent statement in the current paper?

We expect to receive your revised manuscript within two weeks. 

*Published Peer Review History*

*Early Version*

Best wishes,

Roli

Senior Editor,

rroberts@plos.org,

PLOS Biology

DATA POLICY:

I note that your raw data are both third-party and clinically sensitive, which are covered by exemptions under our Data Policy. However, we do ask for the numerical data that underlie the figures and results of your paper be made available in one of the following forms:

Regardless of the method selected, please ensure that you provide the individual numerical values that underlie the summary data displayed in the following figure panels as they are essential for readers to assess your analysis and to reproduce it: Figs 3, 4, 5, 6, 7, 8, S1-S13. NOTE: the numerical data provided should include all replicates AND the way in which the plotted mean and errors were derived (it should not present only the mean/average values).

DATA NOT SHOWN?

REVIEWERS' COMMENTS:

Reviewer #1:

Thank you for submitting the heavily revised analyses for re-review. I still have several reservations with the paper, and the corrected analyses lend support to the idea that these patterns are not specific or unique to psychiatric disorders. The paper also has a number of problems including in drowning in data, making it hard to make sense of the results, and problems with the English language. Overall, I do not think there is strong enough evidence to support the conclusions presented, and the manuscript needs parsing into more carefully articulated ideas. I was not invited to review the original submission and feel the work is not strong enough in its current form. 

Reviewer #2:

It was my suggestion to use total visits as the denominator in a corrected analyses, in order to correct for non-specific seasonal trends in medical visits. Having now seen the results, which are largely described as non-sensical by the authors, I must acknowledge my relative lack of statistical expertise to know whether this approach is valid. This being the case, I would defer to a statistical reviewer who can solve this with much more expertise. I do think non-specific seasonal effects should be controlled for…not sure is this is the right approach. 

Reviewer #3:

[identifies herself as Micaela Elvira Martinez]

Dear Authors and Editors,

I am happy to see that the authors took the time and care to add additional details, methods, and figures to this manuscript. It has greatly strengthened the paper. I particularly like the addition of Fig 2. In its revised form, this manuscript is an important contribution to the study of seasonality in health and disease. Below I have included additional comments to be addressed before publication. 

All the best,

Micaela Martinez

Major comments. 

Fig 1A. I am confused by the first panel that has the trend fit between the healthy and unhealthy population. More description is needed to interpret this plot. 

Please provide information about the data structure for the data sets on gitbub. Currently there is no metadata and the files are not usable outside of the python code. In addition to your github, it would be better to put the manuscript data and code in a dedicated data archive, such as Dryad where it will have a DOI and appropriate metadata. 

The plots of how well the models fit the raw data should be included in the supplement and not just github. 

Minor comments.

line 81. "millions" should be million

Fig 1B. The grey lines for the linear fits are very distracting/overwhelming in the plot. Perhaps make the thick grey lines a bit thinner or make them all more transparent. 

Fig 2. Please define what "In: and Out:" indicate for each stratum, I am assuming it is enrollment and disenrollment date. Also, it is really hard to distinguish the green from the blue time series, perhaps change the green to yellow or another color.

line 175. "out of the ordinary" seems to be subjective/odd phrasing because it is not clear what is the "ordinary" pattern.

lines 323. Saying that your corrected analysis has "nonsensical results" seems subjective and to belittle all of your corrected results. I recommend rephrasing this. You can just say that some of the corrected results are not in line with previously published studies; correcting by all visits may not be optimal because seasonal variation in all visits may not be a good proxy for health seeking behavior, but more of a proxy of the seasonality of particular dominant ailments. 

line 361. You should not use "her" when referring to SAD patients, it indicates they are all female. 

line 390 paragraph. It might be helpful to also discuss here that healthcare coverage differs for these countries and may impact health seeking behavior as well and the diagnosis rate. Without nationalized health care, people with chronic diseases (such as psychiatric disorders) who lack health insurance may be less likely to seek treatment unless it is an emergency or they are having an acute episode. 

Discussion. It would be worth mentioning in the discussion that in the US there are school-based mental health awareness programs. Thus, some of the summertime dip in depression/anxiety diagnosis for 5-18 year-olds could be due in part to underreporting when students are out of school. I don't know much about these programs, but did a quick google scholar search to try and find info. Here is one paper: https://www.ncbi.nlm.nih.gov/pmc/articles/PMC5123790/

---

## [Editor Report · Decision Letter 3]

30 Jun 2021

Dear Andrey,

Thank you for submitting your revised Research Article entitled "Do psychiatric diseases follow annual cyclic seasonality?" for publication in PLOS Biology. 

We're nearly there, but I'm afraid that I have a handful of annoying requests for you to do:

a) Please could you cite the Github and/or Dryad URLs (and potentially the supplementary files) clearly in each relevant main and supplementary Figure legend, e.g. "The data underlying this Figure can be found in https://doi.org/10.5061/dryad.vdncjsxv6"). This may seem repetitive, but it makes the Figs and their legends more standalone.

b) It may be helpful to rename the supplementary data files (currently "S1.zip" and "S1.z01") as "S1_Data" and "S2_Data" as I suspect that my colleagues in the Production department will ask you to do this anyway.

c) Thanks for including the ethics statement - could you possibly incorporate it into the text of the methods section, rather than having it as a separate statement?

d) I note that you told me that you couldn't change the article type to Research Article. I've now done this, so no further action is required.

We expect that these won't take you long, so we expect to receive your revised manuscript within one week. 

*Published Peer Review History*

*Early Version*

Best wishes,

Roli

Senior Editor,

rroberts@plos.org,

PLOS Biology

---

## [Editor Report · Decision Letter 4]

2 Jul 2021

Dear Andrey,

On behalf of my colleagues and the Academic Editor, Marcus Munafò, I'm pleased to say that we can in principle offer to publish your Research Article "Do psychiatric diseases follow annual cyclic seasonality?" in PLOS Biology, provided you address any remaining formatting and reporting issues. These will be detailed in an email that will follow this letter and that you will usually receive within 2-3 business days, during which time no action is required from you. Please note that we will not be able to formally accept your manuscript and schedule it for publication until you have made the required changes.

PRESS: We frequently collaborate with press offices. If your institution or institutions have a press office, please notify them about your upcoming paper at this point, to enable them to help maximise its impact. If the press office is planning to promote your findings, we would be grateful if they could coordinate with biologypress@plos.org. If you have not yet opted out of the early version process, we ask that you notify us immediately of any press plans so that we may do so on your behalf.

Sincerely, 

Roli

Roland G Roberts, PhD 

Senior Editor 

PLOS Biology

rroberts@plos.org